# FROM SKILLS TO PLANS: AUTOMATIC SKILL DISCOVERY AND SYMBOLIC INTERPRETATION FOR COMPOSITIONAL TASKS

## ABSTRACT

Deep Reinforcement Learning (DRL) has struggled with pixel-based controlling tasks that have numerous entities, long sequences, and logical dependencies. Methods using structured representations have shown promise in generalizing to different object entities in manipulation tasks. However, they lack the ability to segment and reuse basic skills. Neuro-symbolic RL excels in handling long sequential decomposable tasks yet heavily relies on expert-designed predicates. To address these challenges, we introduce a novel pixel-based framework that combines entity-centric decision transformers with symbolic planning. Our approach first automatically discovers and learns basic skills through experiences in simple environments without human intervention. Then, we employ a genetic algorithm to enhance these basic skills with symbolic interpretations. Therefore, we convert the complex controlling problem into a planning problem. Taking advantage of symbolic planning and entity-centric skills, our model is inherently interpretable and provides compositional generalizability. The results of the experiments show that our method demonstrates superior performance in long-horizon sequential tasks and real-world object manipulation.

## 1 INTRODUCTION

Deep Reinforcement Learning (DRL) has been successfully applied to various fields, including video games (Mnih, 2013), autonomous driving (Sallab et al., 2017), and robotics (Kober et al., 2013). However, building flexible and adaptive robotic agents that can accomplish a diverse set of tasks in novel and complex environments remains a significant challenge in DRL. Such tasks typically demand the agent to formulate long-term plans for logically dependent goals, requiring it to combine diverse skills in complex scenarios involving multiple objects. A significant challenge of these tasks is the need for *compositional generalization*. We can assess it in terms of two distinct factors: (1) different attributes of objects than in training, and (2) different compositions of goals and their corresponding skills, including variations in logical order (Lin et al., 2023).

To address the above challenge, several methods (Zadaianchuk et al., 2020; 2022; Mambelli et al., 2022; Haramati et al., 2024) incorporate structured representations into the DRL algorithms of decision transformers through object-centric representations (OCR). With a powerfully structured representation, they show certain generalizability on the types and numbers of objects in object manipulation tasks. However, they cannot simultaneously learn diverse skills due to the catastrophic forgetting problem (McCloskey & Cohen, 1989), where the new information can distort the previously learned knowledge. Besides, they cannot segment the learned integrated policy into diverse fundamental units and reform them to achieve new objectives.

On the other hand, some researchers suggest neuro-symbolic approaches that combine planning and DRL. These approaches aim to handle the combinatorial explosion of possible action sequences by providing high-level abstraction and compositing learned skills. Many existing methods (Illanes et al., 2020; Sun et al., 2020; Zhuo et al., 2021; Mao et al., 2023; Silver et al., 2023) employ a top-down structure by specifying symbolic representation for high-level action models and using them to guide the learning of low-level policies. However, these methods can only work with fully observable environment states and carefully hand-engineered predicates. These predefined predicates

hinder the agent's flexibility, thereby restricting its applicability to real-world tasks such as object manipulation.

In this paper, we propose a pixel-based bottom-up framework that combines the idea of entity-centric decision transformers with planning. Our framework is capable of forming a plan that is composed of skills for complex tasks. It learns fundamental skills from scratch by exploring simple environments through DRL algorithms (Li, 2017) without relying much on expert knowledge. Furthermore, our approach uses genetic programming (Ahvanooey et al., 2019) to induce *symbolic interpretation* for those learned skills, including their preconditions and effects. These interpretations provide the agent with a series of fundamental understandings of its learned skills, which is critical for effective planning. During the evaluation, given a novel and composite task, our agent decomposes the task based on its understanding of the task and basic skills, formulating a sequential plan by search algorithms (Abualigah et al., 2021). Finally, the agent executes this plan and uses its skills to generate specific actions to achieve the final goal. We experimentally verify the efficiency and effectiveness of our framework in two domains: Minecraft, a 2D grid-world environment (Andreas et al., 2017) that focuses on long-horizon planning, and IssacGym (Makoviychuk et al., 2021), a simulated tabletop robotic environment that evaluates the agent's capacity to manage complex 3D object manipulation. Experimental results show that our method can schedule the sequence of skills in an appropriate order with symbolic interpretation. Moreover, the flexible combination of skills allows our approach to handle environments with varying object attributes.

We summarize our key contributions below:

- **End-to-End Pixel-Based Controlling.** Compared to the previous work using processed state input, our pixel-based controlling framework can directly leverage raw image data to perform tasks instead of utilizing the actual state provided by the environment.

- **Automatic Skill Discovery.** Compared to previous work, our approach can automatically discover and learn skills from the environment without any guidance of designed high-level symbolic representations in advance, reducing the dependency on expert knowledge.

- **Symbolic Interpretation.** Our approach is inherently interpretable by planning with a sequential symbolic plan composed of learned skills. By constructing each skill's preconditions and effects, our model can infer the specific task of each skill, thus having a comprehensive understanding of the planning and alleviating the curse of dimensionality.

## 2 RELATED WORK

**Object-Centric RL.** Many recent works employed the structured representation in model-free RL (Colas et al., 2019; Zadaianchuk et al., 2022; Mambelli et al., 2022; Zhao et al., 2022; Zhou et al., 2022; Ferraro et al., 2023; Feng & Magliacane, 2024). Among them, methods such as SMORL (Zadaianchuk et al., 2020) and ECRL (Haramati et al., 2024) leverage object-centric representations (Jiang et al., 2019; Francesco et al., 2020; Daniel & Tamar, 2023) in combination with goal-conditioned attention policies to discover and learn useful skills from raw image data. However, they cannot segment learned skills into fundamental units and reform them to solve novel and complex tasks. In this work, we integrate the idea of entity-centric decision transformers into planning, thus giving our method the ability to learn fundamental skills and understand how to compose them when facing long-horizon tasks.

**Neuro-Symbolic RL.** Several works have explored utilizing symbolic methods in DRL to deal with robotic tasks (Belta et al., 2007; Blaes et al., 2019; Illanes et al., 2020; Kokel et al., 2021; Sehgal et al., 2023; Silver et al., 2023; Acharya et al., 2024), including planning domain definition language (Mao et al., 2023), automata (Hasanbeig et al., 2021), Spectrl (Jothimurugan et al., 2021; Žikelić et al., 2024). Despite the success of previous works, they often require either predefined symbolic structures or predefined skills, limiting their compositional generalizability to complex real-world object manipulation. Therefore, we develop a bottom-up framework to automatically discover skills from simple environments and utilize symbolic interpretation to reform these skills for novel and complex tasks.

## 3 Problem Formulation

We focus on the problem of learning a robotic agent from an environment with only fundamental elements that can eventually achieve a novel and complex goal with learned skills. To form the pixel-based control problem as a planning problem and handle it with the combination of skills, we first introduce the concept of **Factored Goal-Augmented Markov Decision Process** (FGAMDP). Then, we propose a novel approach to bridge the gap between MDP and planning.

### 3.1 Factored Goal-Augmented MDP

We start from a single goal-augmented MDP $\langle \mathcal{S}, \mathcal{G}, \mathcal{A}, \mathcal{P}, \mathcal{R}, \gamma \rangle$ (Liu et al., 2022), where $\mathcal{S}$ is the set of state $s$, $\mathcal{G}$ is the set of goal specification $g$, $\mathcal{A}$ is a set of actions $a$ that the agent executes to interact with the environment, $\mathcal{P}$ is the environmental transition model $\mathcal{P} : \mathcal{S} \times \mathcal{A} \to \mathcal{S}$, $\mathcal{R}$ is defined as the set of reward $r(s_t, a_t)$, and $\gamma \in (0, 1]$ is the discount factor for future rewards. Since the task involves manipulating multiple objects, it is natural that we want to separate them apart and accomplish each goal of the corresponding object. Nevertheless, separating the state is not trivial because the state depicted by the image presents a mixture of relevant and irrelevant information. Therefore, we refine and get the structured information of the state, or *entity*, which is defined as:

**Definition 1** (Entity). *We define the set of entities $e$ as $\mathcal{E}$. Given a state $s \in \mathcal{S}$, we then define a mapping from real state to entity $T_e : \mathcal{S} \to \mathcal{E}$, s.t. $\forall s \in \mathcal{S}, \exists e, e = T_e(s)$ as the entity extraction function, where the entity $e = [e_1, e_2, \ldots, e_m] \in \mathbb{R}^{k \times m}$, where $k$ is the dimension of the structured representation, and $m$ is the number of factorization.*

Hence, we factorize the state set $\mathcal{S}$ into the individual state set of each object $\mathcal{S}_i$ as Guestrin et al. (2003): $\mathcal{S} = \mathcal{S}_1 \times \mathcal{S}_2 \times \cdots \times \mathcal{S}_N$. Similarly, the set of goal $\mathcal{G}$ can also be factorized. Thus, we can define FGAMDP:

**Definition 2** (Factored Goal-Augmented MDP). *The tuple $\langle \mathcal{S}, \mathcal{G}, \mathcal{A}, \mathcal{P}_m, \mathcal{R}, \gamma \rangle$ is a Factored Goal-Augmented MDP over a set of states $\mathcal{S}$ if $\mathcal{S}$ can be further factorized.*

Given the initial state $s_0 \in \mathcal{S}$ and goal $g \in \mathcal{G}$, the agent can sample action from a parameterized policy $\pi(a|s, g)$, thus generating a sequence of actions $a_1, a_2, \ldots, a_n$ to achieve the final goal $g$.

### 3.2 From MDP to Planning

The complexity arising from numerous entities makes controlling on MDP challenging. Therefore, we group entities to form meaningful features and use ground operators to convert MDP to planning.

**Definition 3** (Feature). *We define $\mathcal{F}$ as the set of features $f$, aggregating the environmental entities. For every $f = [f_1, f_2, \ldots, f_n] \in \mathbb{R}^n$, we named it as a feature state. We define the aggregation function $T_f : \mathcal{E} \to \mathcal{F}$, s.t. $\forall e \in \mathcal{E}, \exists f, f = T_f(e)$, which maps a certain entity representation to a feature state. We further define $\mathcal{F}_g$ as the set of goal features $f_g$.*

**Definition 4** (Ground Operator (Kokel et al., 2021)). *Let $\mathcal{O}$ denote the set of ground operators $o$ for the planning problem. Each operator $o$ is a tuple $\langle pre(o), eff(o), \beta \rangle$, where $pre(o) \in \mathcal{F}$ is the set of preconditions that should be satisfied before the ground operator executes, $eff(o)$ is the set of effect indicating the feature state change after the ground operator executes, and $\beta \in \mathcal{F}_g$ is the termination state of $\mathcal{O}$.*

With feature states and ground operators, we can define the planning task as tuple $\langle \mathcal{F}, \mathcal{O}, \mathcal{P}_p, \mathcal{F}_g \rangle$, where $\mathcal{P}_p : \mathcal{F} \times \mathcal{O} \to \mathcal{F}$ is the set of feature state transitions. Given an initial feature state $f_0 \in \mathcal{F}$ and a final goal feature state $f_g \in \mathcal{F}_g$, a sequence of grounding operations $\Pi = o_1 \to o_2 \to \cdots \to o_n$, known as a sequential plan, for the task can be formulated when it is possible to sequentially apply the operation starting at $f_0$ and eventually reach the goal state $f_g$.

## 4 Skill with Symbolic Interpretation

We formally defined the mathematical form of the skill here. We group similar operations, forming a skill to address tasks with similar entities. The skill exhibits three key attributes: (1) it serves as the fundamental operational unit for planning, representing a series of actions to achieve a specific

goal; (2) it is endowed with a logical structure comprising preconditions and effects; and (3) it demonstrates adaptability by generating specific control actions based on varying input states.

**Definition 5** (Skill). *We define the skill as a tuple $l(s) = \langle s, o, \pi_l, pre_l(s, f), eff_l(s, f)\rangle$, where $s \in \mathcal{S}$ is the environment state, $o$ is the ground operator, and $\pi_l(a|s) : \mathcal{S} \to \mathcal{A}$ is a specific policy.*

In the above definition, the precondition $pre_l(s, f)$ and effect $eff_l(s, f)$ are both a function of the input state and feature, which means there are multiple legal feature states for a particular skill. Applying the skill to different feature states would have different effects.

For a given task, we define the initial state as $s$ and the goal state as $g$. We can find the initial state in feature representation $f_0 = T_f(T_e(s))$, $f_g = T_f(T_e(g))$. Then we can form a sequential plan $\Pi = l_1(s_1), l_2(s_{t+1}), \ldots, l_n(s_{(n-1)*t+1})$, s.t. $pre_{l_i}(s_{i*t+1}, f_i) = pre_{l_i}(s_{i*t+1}, f_{i-1} + eff_{l_{i-1}}(s_{(i-1)*t+1}, f_{i-1})) = True$. Then finally we have the trace $\tau = s_1 \xrightarrow{a_1} s_2 \xrightarrow{a_2} \ldots \xrightarrow{a_{n*t}} g$, which will achieve the final goal.

## 5 METHOD

Our goal is to design a framework that can automatically discover and learn fundamental skills and form a symbolic plan composed of these skills for complex tasks. The overall structure of our framework is depicted in Figure 1. It mainly consists of three parts: Skill Learning, Symbolic Inductive Inference, and End-to-End Pixel-Based Planning. We will elaborate on these components in the following sections.

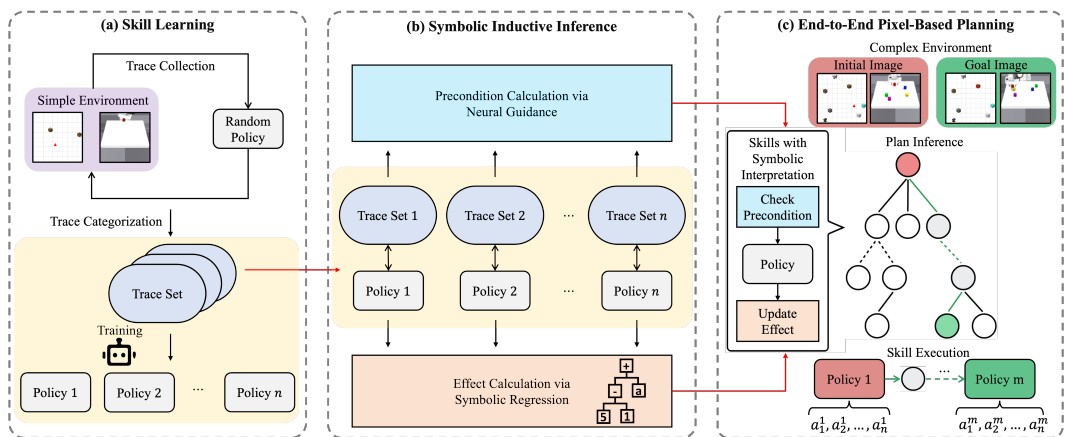

Figure 1: The overview of our framework. (a) **The training stage of the skills.** Traces are collected through random sampling and are then classified into several sets. We train a skill for each set of traces. (b) **The induction of skills.** We endow skills with symbolic interpretations by incorporating neural guidance and symbolic regression to learn preconditions and effects respectively. (c) **End-to-end pixel-based planning.** Given images of the current state and goal, our framework employs MCTS to search for a plan satisfying the precondition and effect of each skill at every stage, from which specific actions are generated to achieve the goal.

### 5.1 FEATURE EXTRACTION

Our objective is to extract a compact and disentangled OCR from raw image observations, capturing most of the essential information. Furthermore, we can aggregate these entities and construct the features of our environment with our feature extractor, preparing for high-level logical operations.

**Entity-Centric Representation.** Given a tuple of raw image observations $(I_1^s, \ldots, I_N^s)$, we process each image separately using a pre-trained Deep Latent Particles (DLP) (Daniel & Tamar, 2023) model, extracting a set of $N$ entities $\{e_n^m\}_{n=1}^N$, $m$ indexing the number of factorization, We denote the entity of raw image observations $I_n^s$ by $e_n^m$. We can also extract and represent the entity of goal observations as $\{g_n^m\}$. Details of DLP can be found in Appendix B.1.

**Entity Aggregation.** As discussed in definition 3, we aggregate entities to features for planning. We develop an aggregation transformer inspired by the entity-centric architecture (Haramati et al., 2024), which processes the OCR entities into the feature state of the environment. The aggregation transformer comprises self-attention (SA) and cross-attention (CA) as its core components. SA is intended to extract *important attributes from the observation* more effectively, while CA is designed to capture the *temporal difference* between current state entities. The set of state entities $\{e_n^m\}_{n=1}^N$ are processed by a sequence of Transformer (Vaswani, 2017) blocks: SA → CA → SA, followed by a MLP (Murtagh, 1991). A detailed architecture is depicted in Figure 7. The aggregation transformer is trained to minimize the mean square loss:

$$\mathcal{L}_{AT}(\hat{\boldsymbol{f}}) = \frac{1}{N} \sum_{i=1}^N \left(\boldsymbol{f}_i - \hat{\boldsymbol{f}}_i\right)^2, \tag{1}$$

where $N$ is the total number of training data. We have the flexibility to define a relatively large number of features that we expect to be beneficial when describing the task.

## 5.2 SKILL LEARNING

We aim to learn the policy $\pi_l$ for a skill $l(\boldsymbol{s}) = \langle \boldsymbol{s}, o, \pi_l, pre_l(\boldsymbol{s}, \boldsymbol{f}), eff_l(\boldsymbol{s}, \boldsymbol{f}) \rangle$ from scratch, which relies on using the collected trace, composed of the original state, as training data. The key idea is that we first collect traces from interaction with the environment, and then we categorize these traces according to the feature change. Finally, we train the agent to learn the skill policy for each collection of traces.

**Trace Generation.** We first collect a significant number of traces using a random policy. Instead of struggling with the complex environments where our agent should work during evaluation, we collect these traces from variant and simple training environments. They represent the basic units of the task, thus usually containing only a few entities and features distilled from the interaction between a single object and the environment.

**Trace Categorization.** After trace generation, we categorize the trace into different sets by their feature, getting the offline dataset based on different feature changes. If changes are observed between two consecutive features, we segment the trace between these two features. We employ a k-means clustering algorithm (Ahmed et al., 2020), with objective function:

$$\arg\min_{\mathcal{T}} \sum_{i=1}^{\mathcal{K}} \frac{1}{|\mathcal{T}_i|} \sum_{\tau_1, \tau_2 \in \mathcal{T}_i} \|\tau_1 - \tau_2\|_2, \tag{2}$$

where $\tau$ is the trace, $\mathcal{T}_i \subseteq \mathcal{T}$ is each classified cluster, and $\mathcal{K}$ is the total number of clusters.

**Training.** It is worth noting that the training algorithm for $\pi_l$ is agnostic of the planning framework. Here we adopt the goal-conditioned behavior cloning (GCBC) algorithm (Lynch et al., 2020) to learn skill policy from categorized offline datasets. The network of the policy is also composed of transformer blocks. The outline of the policy network is a composed structure of SA and CA. It can model the relationship between the current state and the goal. We apply the GCBC loss to train each skill that can achieve the best performance. With a sequence of entity representations $\{E_1, ..., E_T\}$, where $E_i = [\boldsymbol{e}_1, ..., \boldsymbol{e}_N]$, the loss is as following:

$$\mathcal{L}_{GCBC} = -\|E_t - E_g\|_2 - \frac{\beta}{T} \sum_{t=1}^T \log(\pi_l(\boldsymbol{a}_t | E_t, E_g)), \tag{3}$$

where $\beta$ is the hyperparameter which can be adjusted. The first term represents the goal-conditioned loss, and the second on is the GCBC loss.

## 5.3 SYMBOLIC INDUCTIVE INFERENCE

To form a plan using the skills, we need to get a symbolic interpretation of the skill. In our skills, symbolic interpretation is the precondition and effect of the skills. The precondition and effect are presented in the form of a mathematical formula, which uses arithmetic operation $\{+, -, \times, \div, >\}$ as the operation set; that is, we want to form all the preconditions and effects as a polynomial.

**Symbolic Regression.** Given a raw skill policy $\pi_l$, the induction module proceeds to search for the effect $eff_l(\boldsymbol{s}, \boldsymbol{f})$ for this skill policy. As mentioned in section 4, the effect might change as the input state changes. Here, we assume the state change can be reflected by the feature $\boldsymbol{f}$. Then we have $\boldsymbol{f}'_{final} = eff(\boldsymbol{f}_{init})$, which can be formulated as a symbolic regression problem.

For symbolic regression, we use the PySR (Cranmer, 2023), which is a multi-population evolutionary algorithm. PySR can perform feature selection and select the most significant features from the feature vector $\boldsymbol{f}$ by providing a user-defined number of features. Moreover, it also supports customizing the operator and the loss function. Here, we design an element-wise loss function:

$$\mathcal{L}_{SR} = \|\boldsymbol{f}_{pred} - \boldsymbol{f}_{target}\|_2 + complexity, \tag{4}$$

where $\boldsymbol{f}_{pred}$ is the prediction result and $\boldsymbol{f}_{target}$ is the ground truth. Here, we introduce a normalization term complexity to prioritize the effect function using a simple mathematical format.

**Precondition Rule.** Since the features in the environment might be complicated, determining whether a skill can be applied in the current stage is challenging. Here, we apply a neural guidance approach to find the symbolic precondition rules for a skill. We first learn a neural network that takes in the feature vector $\boldsymbol{f}$ and outputs a boolean result. Then, we try to use an EQL network (Sahoo et al., 2018) with a unary activation function to simulate the behavior of the neural network. The EQL network is optimized by the distribution loss:

$$\mathcal{L}_{EQL} = -\left| \log \frac{\pi_{EQL}(p_{pos}|\boldsymbol{f})}{\pi_{nn}(p_{pos}|\boldsymbol{f})} \right| - \left| \log \frac{\pi_{EQL}(p_{neg}|\boldsymbol{f})}{\pi_{nn}(p_{neg}|\boldsymbol{f})} \right|, \tag{5}$$

where $p_{pos}$ is the possibility of the positive result and $p_{neg}$ is the possibility of the negative result.

**Dependency Graph.** As we have precondition and effect for skill, we can construct a dependency graph through topological generation (Manber, 1989). The preconditions are some inequations of the skill, such as $wood \geq 0$. The effects are some functions that update the feature, such as $wood+1$. We define all the ground operators as vertex $v$ of a graph. For $\forall v_i, v_j \in V, v_j \neq v_j$, there is a directed edge $e_{ij}$ between $v_i, v_j$, iff $pre_{l_j}(\boldsymbol{s}, eff_{l_i}(\boldsymbol{s}, \boldsymbol{f}_0)) = True$, we denote this relation as $v_i \prec v_j$, where $\boldsymbol{f}_0$ is an empty feature vector, and construct a directed graph $G$.

## 5.4 END-TO-END PIXEL-BASED PLANNING

In this section, we introduce the overall process where our framework generates detailed actions given an initial image and a goal image.

**Subgoal Image Generation.** Given an image of current state $\boldsymbol{s}$ and goal state $\boldsymbol{g}$, our feature extraction module can convert the pixel input into feature representation $\boldsymbol{f}_{init}$ and $\boldsymbol{f}_g$. Additionally, we use an image segmentation algorithm to segment the image according to its semantics, thus forming state $\boldsymbol{s}^i$ and subgoal state $\boldsymbol{g}^i$.

**Plan Inference.** This part focuses on generating a skill composition that can fit the goal feature $\boldsymbol{f}_g$ and the input feature $\boldsymbol{f}_{init}$. We use Monte Carol Tree Search (MCTS) (Świechowski et al., 2023) as the search algorithm. We define the $\mathrm{Importance}$ of each ground operator by its depth $d$, in-degree $l^{in}$, out-degree $l^{out}$ in the dependency graph, the number of positive feature $\boldsymbol{f}_p$ and the number of operators calls $c$:

$$\mathrm{Importance}(l) = \frac{d + \ln l^{in} + \ln l^{out} + \alpha}{\min(c, \boldsymbol{f}_p) + 1}, \tag{6}$$

where $\alpha$ represents the hyperparameter of decay of function calls. With the $\mathrm{Importance}$ above, we apply the Upper Confidence Bound to Trees (UCT):

$$\mathrm{UCT}(l) = \mathrm{Importance}(l) + \sqrt{C \times \log \frac{\mathrm{visited}(\mathrm{parent}(l))}{\mathrm{visited}(l)}}, \tag{7}$$

where $\mathrm{Importance}$ is the reward of taking skill $l$, $C$ is a hyperparameter to balance the exploration and exploitation, $\mathrm{visited}$ is a function to get the number of accessed times of skill $v$, and $\mathrm{parent}$ is to get the previous skill of the skill $v$. We select the next skill with the max UCT, $\max_l UCT(l)$. Then, we expand this skill if there are untried skills. Finally, we simulate some steps and update the $\boldsymbol{f}_p$ of each node according to the reward, the number of visits, and UCT.

**Skill Execution.** For each skill, we have a pixel-based input state $s^i$, which represents the current state, and a subgoal image $g^i$. And we set the policy a time horizon as $t$. A skill can execute for consecutive $t$ timesteps before switching to the next one. During the execution, the image segmentation model segments the subgoal image for the skills, and then the skills take in the subgoal figure $\hat{s}^i$ and output an action $\pi_l(a|\hat{s}^i, g^i)$. Thus, we find an approach to accomplish the whole task.

## 6 EXPERIMENTS

To evaluate the performance of our model, we select two different types of environments. One is the Minecraft environment, which verifies a series of long-horizon compositional tasks. The other is IsaacGym, a robotic arm simulation environment employed to assess the performance of compositional generalization tasks.

**Environments.** Minecraft is an $n \times n$ grid world environment. It is inspired by the computer game Minecraft and is similar to the environment in previous works (Brooks et al., 2021; Hasanbeig et al., 2021; Kokel et al., 2021; Liu et al., 2024). An agent can move along four directions $\{\text{up}, \text{down}, \text{left}, \text{right}\}$ and interact with objects with learned skills. Different from the previous environment, our inputs are image maps with different objects in the map. The tasks are as follows:
`Make-Stick`: A basic experiment that urges the agent to produce a stick.
`Make-Mass-Sticks`: Produce a huge number of sticks that require **repeating the same skills many times**.
`Pickup-Iron`: Make several tools and leverage them to pick up iron.
`Multiple-Goals`: Collect four items in an inherent order.
`Make-Enhance-Table`: The most difficult **long-horizon** task that requires the cooperation of many skills to accomplish.

IsaacGym (Makoviychuk et al., 2021) is a simulated tabletop robotic object manipulation environment. The environment includes a robotic arm set in front of a table with various cubes and buttons in different colors. The agent observes the system's state through visual input and performs actions in the form of deltas in the end effector coordinates $a = (\Delta x, \Delta y, \Delta z, \Delta g)$, where $\Delta g$ indicates whether the gripper is open or close. At the beginning of each episode, both the current cube positions and the goal positions are randomly initialized on the table. The tasks are as follows:
`Push`: Push cubes with **randomized numbers and color** to the goal location.
`Push-Grab-Lift`: Manipulate cubes of randomized numbers and color to their goal positions by **pushing and lifting** operation.
`Ordered-Press`: Press different buttons **in an inherent order**.

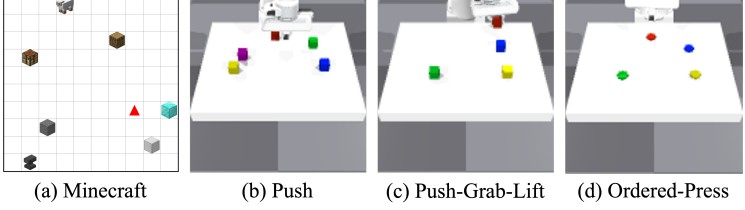

(a) Minecraft      (b) Push      (c) Push-Grab-Lift      (d) Ordered-Press

Figure 2: The environments used for experiments in this work.

**Baselines.** We extensively compare our framework to various DRL algorithms with pixel-based decision transformers (ECRL, SMORL) learning from *rewards*, imitation learning methods (GAIL) learning from *demonstration*, and methods that combine planning and DRL (Deepsynth, DiRL).

- **SMORL** (Zadaianchuk et al., 2020) adopts object-centric representations in combination with goal-conditioned attention policies to discover and learn useful skills.
- **ECRL** (Haramati et al., 2024) uses object-centric representations with the entity-interaction transformer to discover and learn useful skills.
- **GAIL** (Ho & Ermon, 2016) mimics expert behaviors via learning a generative adversarial network whose generator is a policy.

| Task | Make-Stick | Make-Mass-Sticks | Pickup-Iron | Multiple-Goals | Make-Enhance-Table |
|------|------------|------------------|-------------|----------------|--------------------|
| SMORL | 0.511 | 0.374 | 0.332 | 0.281 | 0.112 |
| ECRL | 0.462 | 0.358 | 0.371 | 0.297 | 0.187 |
| GAIL | 0.513 | 0.469 | 0.415 | 0.302 | 0.164 |
| DiRL | 0.867 | 0.863 | 0.794 | 0.719 | 0.581 |
| DeepSynth | 0.901 | 0.853 | 0.821 | 0.693 | 0.557 |
| **Ours** | **0.958** | **0.938** | **0.917** | **0.887** | **0.750** |

Table 1: Success rate of Minecraft.

- **DeepSynth** (Hasanbeig et al., 2021) uses an automaton to find the substructure of tasks and execute the subtasks using the low-level controller.
- **DiRL** (Jothimurugan et al., 2021) uses a predefined logical specification to decompose tasks into subtasks and then solve subtasks by DRL controller.

## 6.1 Long-Horizon Sequential Task

We evaluate the different methods in the Minecraft Environment to test the performance in some long-horizon tasks. Results are presented in Table 1. For long horizon planning tasks, **ECRL, SMORL** achieves a low success rate because they cannot handle temporal logic tasks. **Deepsynth** uses an automaton-based high-level structure for task decomposition, so it has a relatively high success rate in simple tasks. However, as the tasks become complex, their performance drops sharply since the search space for the automaton is too big for the algorithm to cover.

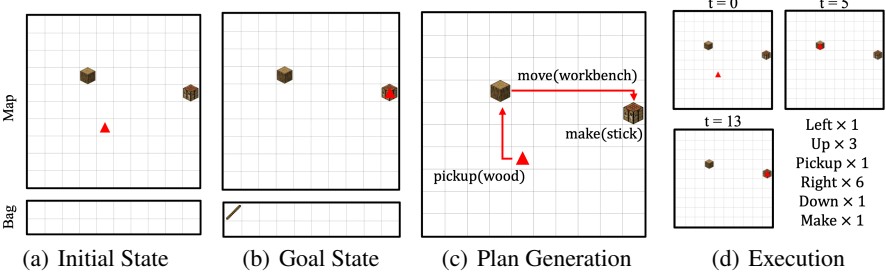

(a) Initial State     (b) Goal State     (c) Plan Generation     (d) Execution

Figure 3: End-to-end pixel-based planning of `Make-Stick`.

## 6.2 Object Manipulation in Real-World Environment

We evaluate different methods in the IsaacGy and present results in Table 2. Here, we mainly demonstrate the overall success rate and the success fraction. For a single cube, the success rate equals the success fraction. For `Push`, we observe that most of the structured baselines **ECRL, SMORL** can achieve a high success rate. Contrarily, conventional behavior cloning **GAIL** performs poorly as the number of cubes increases because of its poor compositional generalizability. **Deepsynth** and **DiRL** uses the idea of task decomposition, however, the decompositional logic is simple and relies on expert knowledge. Thus, they also perform poorly as the number of cubes increases. `Push-Grab-Lift` and `Ordered-Press` have some logical dependency on their subtasks, the performance of **ECRL, SMORL** is much poorer than our model because these two models have no awareness of the temporal attributes of sub-tasks, which shows the superiority of our skills with symbolic interpretation. Other detailed results are in Appendix D.2.

## 6.3 Symbolic Interpretation

Symbolic interpretation is an essential feature of our skills, which enables the searching algorithm to find a feasible plan for a complex task. We have shown the symbolic interpretation of IsaacGym

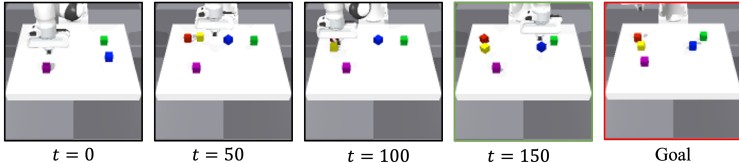

| | $t = 0$ | $t = 50$ | $t = 100$ | $t = 150$ | Goal |

Figure 4: A sample trace of an agent in real-world object manipulation `Push`.

| | Cubes | 1 | 2 | 3 | 4 | 5 |
|---|---|---|---|---|---|---|
| | SMORL | 0.990 / 0.990 | 0.838 / 0.911 | 0.509 / 0.794 | 0.438 / 0.722 | 0.302 / 0.618 |
| | ECRL | 0.973 / 0.973 | 0.963 / 0.981 | 0.838 / 0.942 | 0.723 / 0.912 | 0.570 /0.876 |
| Push | GAIL | 0.955 / 0.955 | 0.750 / 0.875 | 0.478 / 0.706 | 0.438 / 0.697 | 0.396 / 0.646 |
| | DiRL | 0.935 / 0.935 | 0.873 / 0.912 | 0.741 / 0.797 | 0.612 / 0.753 | 0.505 / 0.687 |
| | DeepSynth | 0.919 / 0.919 | 0.861 / 0.883 | 0.803 / 0.877 | 0.635 / 0.720 | 0.493 / 0.579 |
| | **Ours** | **1.000 / 1.000** | **1.000 / 1.000** | **0.875 / 0.958** | **0.750 / 0.922** | **0.688 / 0.900** |
| | SMORL | 0.250 / 0.719 | 0 / 0.625 | 0 / 0.469 | 0 / 0.375 | 0 / 0.188 |
| Push- | ECRL | 0.250 / 0.797 | 0 / 0.538 | 0 / 0.438 | 0 / 0.375 | 0 / 0.250 |
| Grab- | GAIL | 0.488 / 0.670 | 0.348 / 0.532 | 0.101 / 0.329 | 0.031 / 0.281 | 0.010 / 0.157 |
| Lift | DiRL | 0.521 / 0.674 | 0.355 / 0.578 | 0.235 / 0.416 | 0.065 / 0.343 | 0.025 / 0.188 |
| | DeepSynth | 0.502 / 0.755 | 0.330 / 0.581 | 0.187 / 0.453 | 0.083 / 0.302 | 0 / 0.009 |
| | **Ours** | **0.625 / 0.875** | **0.500 / 0.828** | **0.500 / 0.863** | **0.156 / 0.734** | **0.125 / 0.643** |
| | SMORL | 0.980 / 0.980 | 0.686 / 0.877 | 0.513 / 0.797 | 0.372 / 0.701 | 0.158 / 0.629 |
| | ECRL | **1.000 / 1.000** | 0.625 / 0.913 | 0.427 / 0.835 | 0.354 / 0.801 | 0.277 / 0.778 |
| Ordered- | GAIL | 0.971 / 0.971 | 0.862 / 0.905 | 0.697 / 0.764 | 0.535 / 0.712 | 0.328 / 0.567 |
| Press | DiRL | 0.982 / 0.982 | 0.931 / 0.955 | 0.841 / 0.862 | 0.703 / 0.826 | 0.662 / 0.691 |
| | DeepSynth | 0.925 / 0.925 | 0.908 / 0.934 | 0.803 / 0.879 | 0.655 / 0.784 | 0.535 / 0.681 |
| | **Ours** | 0.990 / 0.990 | **0.938 / 0.969** | **0.875 / 0.958** | **0.813 / 0.953** | **0.813 / 0.950** |

Table 2: Success rate and success fraction of real-world object manipulation.

in Table 6 and Minecraft in Table 5, where the preconditions are the boolean formula and the effects are in the format of a function.

Taking `Make-Stick` as an example, the agent first compares the initial images and goal images demonstrating its task and is aware that it should make a stick at the workbench as shown in Figure 5. The preconditions of the last action make(stick) are wood $\geq 1$ and at_workbench $= 1$, which means the agent should move to the workbench with a wood. The effect of move(workbench) is at_workbench $= 1$, thus we have move(workbench) $\prec$ make(stick). Similarly, the effect for pickup(wood) is wood $+ 1$, thus we have pickup(wood) $\prec$ make(stick). We can also get this relation from the dependency graph in Figure 9. In the plan generation module, the agent induces a symbolic plan by using MCTS, forming a sequence: pickup(wood) $\rightarrow$ move(workbench) $\rightarrow$ make(stick). This sequence satisfies the aforementioned partial order relation and can accomplish the task, thus forming our final plan.

## 6.4 COMPOSITIONAL GENERALIZATION

This section investigates our method's ability to achieve zero-shot compositional generalization. Specifically, we hope the agent can apply learned skills to objects with similar features. Hence, we present some inference scenarios requiring compositional generalization. Additional results are in Appendix D.3.

**Different Color and Shape of Objects in IsaacGym.** In the object manipulation environment Isaacgym, our skills are trained with a single red cube. We try to eliminate the influence of color and shapes on the skills by adding noise to the related entities output by the OCR. At the testing stage, our model can operate on objects with random shapes and colors. The shapes can be chosen from {cuboid, cylinder, star}. We test our model by the `Push` tasks and set the number of objects as three. From the demonstration in Figure 6(a), we can see that our model generalizes to the objects

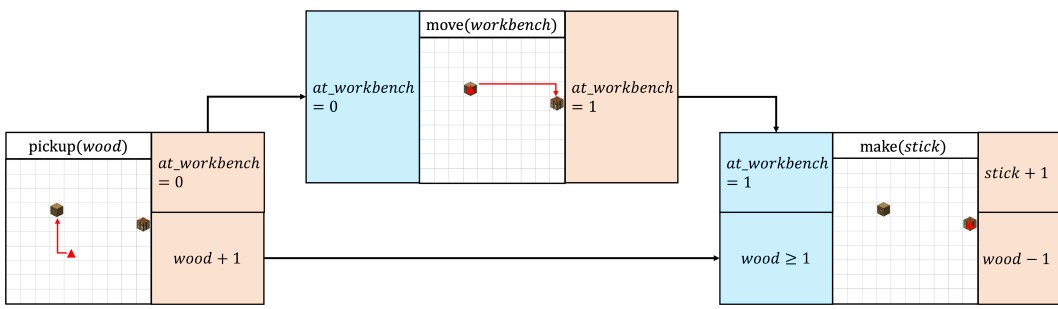

Figure 5: Skill relation based on symbolic interpretation in `Make-Stick`. Expressions in the blue box represent the *precondition*, and orange denotes the *effect*.

with unseen entities and maintains a high success rate in push tasks. Detailed results are in Appendix Table 11.

**Different Craft tasks in Minecraft.** In Minecraft, skills are trained in an environment with elements $\{\text{wood}, \text{stone}\}$. We also expect our skills, such as *pick* and *make*, can be generalized into crafting objects with similar entity **without further training**. In the evaluation environment, we introduce new objects $\{\text{grass}, \text{bamboo}\}$ and the corresponding crafting tasks. We test our model on some collecting and crafting tasks by substituting the materials in the map. The result is demonstrated in Figure 6(b). Our model maintains the success rate on these similar tasks. The complete results are in Appendix Table 12.

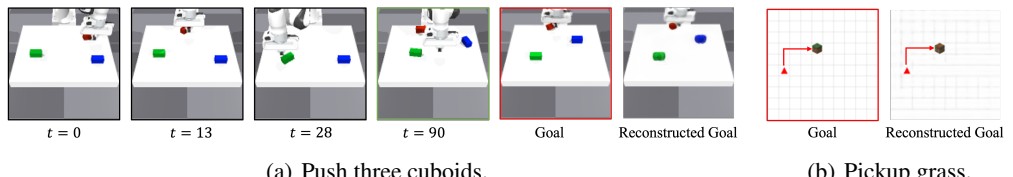

(a) Push three cuboids.        (b) Pickup grass.

Figure 6: Rollout of an agent train on some objects, then provide goal image with objects of slightly different entities. The agent finishes the new tasks without additional training, which shows the ability to perform compositional generalization.

## 7 CONCLUSION

We present a model combining a planning framework with DRL to solve pixel-based control challenges. Our model can autonomously acquire basic skills through interaction with the environment, minimizing the need for expert knowledge. Moreover, by providing symbolic interpretations for skills, we can form a sequential plan for long-horizon tasks through search algorithms such as MCTS. Additionally, our model leverages composable skills and a transformer-based action policy, which provides compositional generalizability to tasks that share similar features. Our model has shown great performance on long-horizon, pixel-based control problems based on this superiority.

**Limitation.** Our model requires a pre-trained image segmentation model. While basic image segmentation models have achieved promising results in our experiments, more complex tasks may require further advancements and refinements in image semantic segmentation techniques. Additionally, the approach using discrete features as an interface may induce some inaccuracy and inflexibility. Some states with slight differences may share the same feature representation.

**Future Work.** For future work, one interesting direction is to explore more advanced ways for automatic skills generation. Currently, the skill generation relies on classifying the collected traces. We can further improve it with reward-based or entropy-based methods in the future. Another possible direction is to employ generative models, such as diffusion models, to replace the current image segmentation approach to generate sub-goal images.

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

APPENDIX

## A    ALGORITHM

We outline the algorithm of our end-to-end pixel-based framework below. Lines 2 to 15 detail the preprocessing of traces, involving categorizing them into distinct groups for subsequent training. Lines 16 to 24 describe the process of skill formation through symbolic interpretation of the traces. Lines 25 to the end encompass the planning and execution of different tasks.

---

**Algorithm 1:** The whole training and evaluation of the framework.

---

**Input:** The total trace collecting step $N$. The evaluation task $\mathbb{T}_{eval}$.

1  Randomly initialize some simple environment $e_i$;
2  **for** $j \leftarrow 0$ **to** $N$ **do**
3      | Interact with simple environment with random policy $\pi_{random}$;
4      | Collect the trace $\tau_{ori}$;
5  **end**
6  **for** $\tau_{ori} \in T$ **do**
7      | $p \leftarrow 0, q \leftarrow 0$;
8      | **while** $T_f(T_e(\boldsymbol{s}_p)) = T_f(T_e(\boldsymbol{s}_q))$ **do**
9      |     | $q$++;
10     | **end**
11     | Segment the trace $\tau_{ori}[p:q]$;
12     | Insert the trace to a trace set $\mathcal{S}_{trace}$;
13     | p = q;
14 **end**
15 Classify the trace using cluster algorithm;
16 **for** *trace cluster* $\mathcal{S}_i \in \mathcal{S}_{trace}$ **do**
17     | Randomly initialize policy $\pi_i$;
18     | Training policy $\pi_i$ with GCBC algorithm;
19 **end**
20 **for** $i \leftarrow 0$ **to** $|\mathcal{S}_{trace}|$ **do**
21     | Find the effect of $\pi_i$ through PySR;
22     | Find the precondition of $\pi_i$ through neural guidance algorithm;
23     | Form a skill with symbolic interpretation $l(\boldsymbol{s}) = \langle \boldsymbol{s}, o, \pi_l, pre_l(\boldsymbol{s}, \boldsymbol{f}), e\!f\!f_l(\boldsymbol{s}, \boldsymbol{f}) \rangle$;
24 **end**
25 Get the initial state $\boldsymbol{s}_{init}$ and goal state $\boldsymbol{g}$ of $\mathbb{T}_{eval}$;
26 Get the initial feature $\boldsymbol{f}_{init}$ and goal feature $\boldsymbol{f}_g$;
27 Using MCTS to find a path $l_1 \rightarrow l_2 \rightarrow \cdots \rightarrow l_n$ from $\boldsymbol{f}_{init}$ to $\boldsymbol{f}_g$;
28 **for** $j \leftarrow 0$ **to** $n$ **do**
29     | **for** $t \leftarrow 0$ **to** *volley* **do**
30     |     | Get the action $\boldsymbol{a} = \pi_i(\boldsymbol{s}_{j \times volley + t})$ ;
31     |     | Interact with the environment $\boldsymbol{s}_{j \times volley + t + 1} = \mathbb{T}_{eval}(\boldsymbol{a})$;
32     | **end**
33 **end**

---

## B    IMPLEMENTATION DETAILS

### B.1    PRE-TRAINED MODELS

**Object-Centric Representation.**    Our OCR algorithm is based on the DLP algorithm. DLP (Daniel & Tamar, 2023) is an unsupervised object-centric model for images based on variational autoencoder (VAE) (Kingma, 2013). It provides the latent representation for all the particles.

The foreground representation $e = [e_c, e_s, e_d, e_t, e_f] \in \mathbb{R}^{11}$ is a disentangled latent variable including the following learned attributes: spatial coordinate $e_c \in \mathbb{R}^3$, scale $e_s \in \mathbb{R}^2$, depth $e_d \in \mathbb{R}$, transparency $e_t \in \mathbb{R}$, and visual features $e_f \in \mathbb{R}^4$. Here we set the number of entities as 24.

**Aggregation Function.** We design an aggregation transformer inspired by entity-centric architecture (Haramati et al., 2024), which processes the OCR entities into the feature state of the environment. An architecture outline is presented in Figure 7. The aggregation transformer comprises self-attention (SA) and cross-attention (CA) as its core components. The self-attention tries to grab the relation of entities in a single object. It transforms the input vector, grouping all the entities of a single object. At the CA layer, the transformer network tries to figure out the relation between different objects. After passing the SA and CA network, we let the model pass another (SA) network again. This network considers the result from the previous steps and forms these two steps together, placing self-attention calculation on the overall computing result. Finally, we get the output result, which is a feature vector.

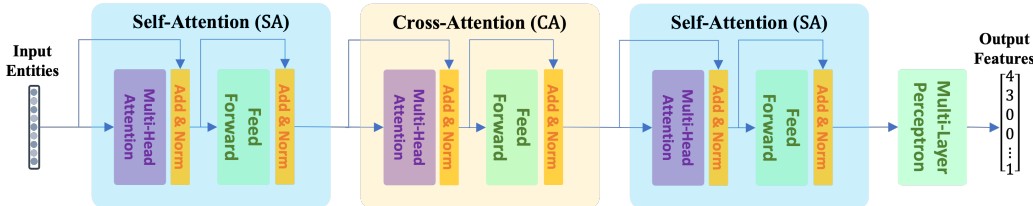

Figure 7: The architecture of the aggregation transformer.

### B.2 Skill Training

It is worth mentioning that our framework is agnostic of the skill policy. We have tried several RL algorithms and finally chose GCBC (Lynch et al., 2020) since it has the best performance. We use GCBC to train the policy $\pi_l$ for the skill. This method extracts goal-conditioned policies using self-supervision on top of raw unlabeled data.

As mentioned in the section 5.2, we collect traces from interaction with the simple environment. Taking the IssacGym environment as an example, we set the object number to be one, the object type to be a cube, and the object color to be red. The agent can operate its gripper to interact with the only object that appeared on the table. Thus, it can collect a tremendous amount of data.

Figure 8 shows the learning curves of the skills, the y-axis is the success rate during the training process. The names and functions of skills are not specified in advance. We name the skill according to its effect.

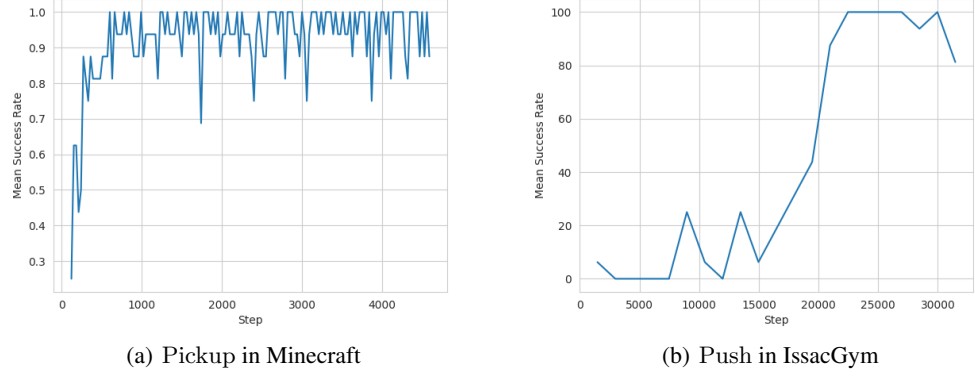

(a) Pickup in Minecraft       (b) Push in IssacGym

Figure 8: The curves of mean success rate during the skill learning process under 96 random goals in IssacGym and Minecraft environments. Notice that the names and functions of skills are not specified in advance.

### B.3 EFFECTS OF SKILL

We use PySR (Cranmer, 2023) for the implementation of the symbolic regression part of the skills, generating the mathematical form of effect. In PySR, we can use specific parameters to control the generation of the formula. The parameter settings for the regressor are in Table 3.

Here we assume that all the effects of skills on features can be characterized by some polynomial expression. Then, we use the binary operation and the constants to form such a relation. We treat the initial feature and final feature of a sequence as the input and output of a function. Then we fit the relationship between tuples of input and output.

| Parameters | Value |
|---|---|
| Number of Iterations | 40 |
| Complexity | 5 |
| Binary Operators | $\{+, -, \times, \div\}$ |
| Unary Operator | $\{>\}$ |
| fractionReplaced | 0.1 |
| shouldOptimizeConstants | True |
| maxsize | 20 |
| procs | 4 |

Table 3: The Parameter Setting of PySR

### B.4 PRECONDITIONS OF SKILL

The precondition is a little different from the effect. The precondition is composed of some boolean expressions. It maps from a feature vector to a boolean output, indicating whether a skill can be adopted under the current stage.

Here we use a similar method as Luo et al. to find the preconditions. We use an EQL network with depth 2. The EQL network takes in an input vector $h$ and output $g = Wh + b$ in each layer, where $W$ and $b$ are learnable parameters. Its activation function in the last layer can be customized according to our needs. For the activation function, we choose the function listed in Table 4. The last three functions are specifically designed for the boolean output since we expect our output to be a boolean. Since the EQL network is shallow and the activation function is given, we can extract the final preconditions to some simple expressions, which are listed in Table 5 and Table 6.

| Parameter | Value |
|---|---|
| Activation Function 1 | $x$ |
| Activation Function 2 | $c$ |
| Activation Function 3 | $x^2$ |
| Activation Function 4 | $2x$ |
| Activation Function 5 | $x > 0$ |
| Activation Function 6 | $x < 0$ |
| Activation Function 7 | $x = c$ |

Table 4: The Activation Function for Learning Precondition

## C  TEST ENVIRONMENTS

### C.1  BASELINE REIMPLEMENTATION

**SMORL.**  We reimplemented SMORL (Zadaianchuk et al., 2020), substituting its original visual model SCALOR (Jiang et al., 2019) with DLP. Additionally, the low-level controller within the SMORL framework was replaced with the same controller used in our proposed method.

**ECRL.**  The original version of ECRL was used directly in our experiments.

**GAIL.**  GAIL (Ho & Ermon, 2016) was reimplemented with several modifications. We integrated DLP to process image input and replaced the actor with the same controller used in our framework. Furthermore, the critic and discriminator networks within GAIL were updated to employ a transformer architecture.

**DeepSynth.**  We reimplement DeepSynth (Hasanbeig et al., 2021), with the original image segmentation algorithm replaced by DLP. We directly implement the automaton synthesis algorithm based on the DLP result. For the low-level controller in DeepSynth, we also use the same controller as our framework to substitute the controller in DeepSynth to ensure a fair comparison.

**DiRL.**  DiRL (Jothimurugan et al., 2021) was reimplemented as a baseline model, incorporating domain-specific knowledge. Rules such as "pick after push" and "pick up wood before going to the craft table" were established to provide high-level guidance for the low-level policy. The policy within DiRL was also replaced with the same controller used in our framework for a more credible comparison.

### C.2  MINECRAFT

We design the features to extract as follows:

- at_wood: A boolean variable representing whether the agent's position is at wood.
- at_stone: A boolean variable representing whether the agent's position is at stone.
- at_iron: A boolean variable representing whether the agent's position is at iron.
- at_gem: A boolean variable representing whether the agent's position is at gem.
- at_sheep: A boolean variable representing whether the agent's position is at sheep block.
- at_workbench: A boolean variable representing whether the agent's position is at the workbench.
- at_toolshed: A boolean variable representing whether the agent's position is at the toolshed.
- wood: The number of wood in the agent's bag.
- stone: The number of stones in the agent's bag.
- iron: The number of iron in the agent's bag.
- gem: The number of gems in the agent's bag.
- stick: The number of sticks in the agent's bag.
- stone_pickaxe: The number of stone pickaxes in the agent's bag.
- iron_pickaxe: The number of iron pickaxes in the agent's bag.
- scissors: The number of scissors in the agent's bag.
- paper: The number of paper in the agent's bag.
- wool: The number of wool in the agent's bag.
- enhance_table: The number of enhanced tables in the agent's bag.
- bed: The number of beds in the agent's bag.
- jukebox: The number of jukeboxes in the agent's bag.

As we have form skills with symbolic interpretation, we can use a graph to describe the dependency relation between different skills. A detailed dependency graph of all the skills in Minecraft is shown in Figure 9.

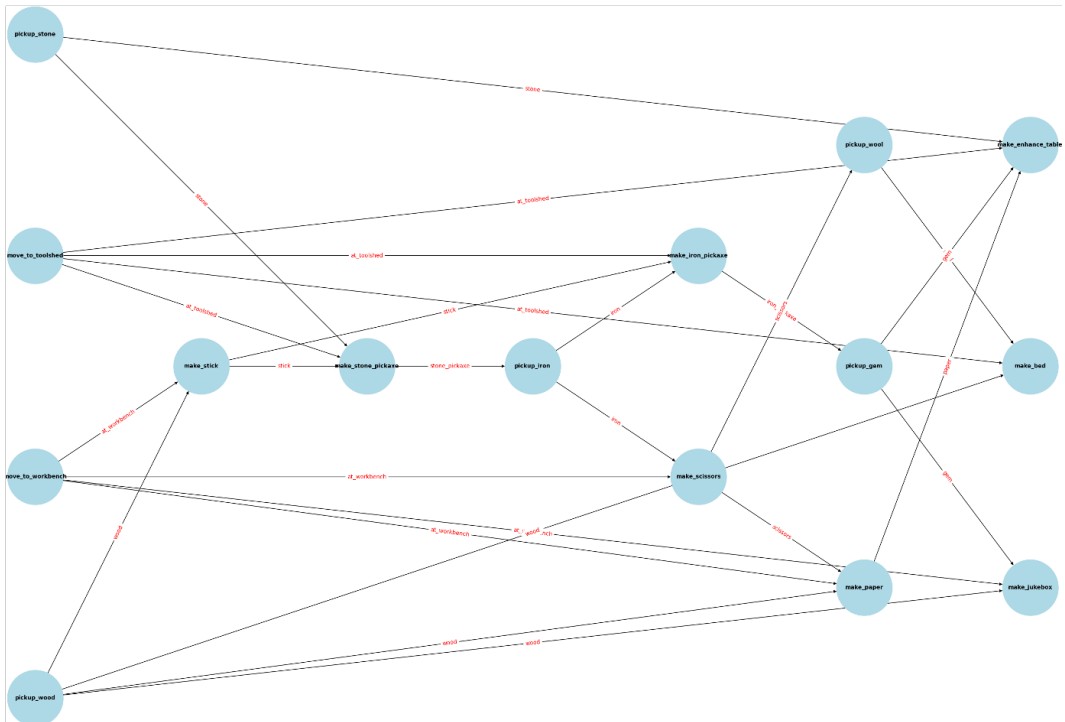

Figure 9: Dependency graph of Minecraft.

## C.3 IssacGym

We design the features of IssacGym to extract as follows:

- num_objects: The number of objects on the table captured by cameras that provide front view and side view.
- xy_goal: The number of objects reaching their goals on the table.
- z_goal: The number of objects reaching their goals in the air lifted by the gripper.
- is_grab: A boolean variable representing whether the gripper grabs the object.
- color_1, ..., color_5: A boolean variable representing whether a color exists. It can record at most five colors.
- next_color: It is an integer that stands for the next color that should be controlled. It guarantees the ordered operation of objects following the color sequence, which is red, green, blue, yellow, and purple in our case.

## C.4 Evaluation Metrics

We mainly evaluate the performance of different methods based on the success rate and success fraction. Apart from these two results, we also record more detailed information in the experiment, including color success, color success fraction, action success, and action success fraction.

- **Success Rate.** The success rate describes the final success rate of the whole task.
- **Success Fraction.** The success fraction is the portion of accomplished subtasks, so this metric is usually higher than the success.

| Skill | Object | Preconditions | Effects |
|---|---|---|---|
| move | workbench | $\text{at\_workbench} = 0$ | $\text{at\_wood} = 0, \text{at\_stone} = 0,$ $\text{at\_iron} = 0, \text{at\_gem} = 0,$ $\text{at\_wool} = 0, \text{at\_workbench} = 1,$ $\text{at\_toolshed} = 0$ |
| | toolshed | $\text{at\_toolshed} = 0$ | $\text{at\_wood} = 0, \text{at\_stone} = 0,$ $\text{at\_iron} = 0, \text{at\_gem} = 0,$ $\text{at\_wool} = 0, \text{at\_workbench} = 0,$ $\text{at\_toolshed} = 1$ |
| pickup | wood | $\text{at\_wood} = 1$ | $\text{wood} + 1$ |
| | stone | $\text{at\_stone} = 1$ | $\text{stone} + 1$ |
| | grass | $\text{at\_grass} = 1$ | $\text{grass} + 1$ |
| | bamboo | $\text{at\_bamboo} = 1$ | $\text{bamboo} + 1$ |
| | iron | $\text{at\_iron} = 1, \text{stone\_pickaxe} \geq 1$ | $\text{iron} + 1$ |
| | gem | $\text{at\_gem} = 1, \text{iron\_pickaxe} \geq 1$ | $\text{gem} + 1$ |
| | wool | $\text{at\_wool} = 1, \text{scissors} \geq 1$ | $\text{wool} + 1$ |
| make | stick | $\text{at\_workbench} = 1, \text{wood} \geq 1$ | $\text{stick} + 1, \text{wood} - 1$ |
| | grass_stack | $\text{at\_workbench} = 1, \text{grass} \geq 1$ | $\text{grass\_stack} + 1, \text{grass} - 1$ |
| | bamboo_fence | $\text{at\_workbench} = 1, \text{bamboo} \geq 1$ | $\text{bamboo\_fence} + 1, \text{bamboo} - 1$ |
| | stone_pickaxe | $\text{at\_toolshed} = 1,$ $\text{stick} \geq 2, \text{stone} \geq 3$ | $\text{stone\_pickaxe} + 1,$ $\text{stick} - 2, \text{stone} - 3$ |
| | iron_pickaxe | $\text{at\_toolshed} = 1,$ $\text{stick} \geq 2, \text{iron} \geq 3$ | $\text{iron\_pickaxe} + 1,$ $\text{stick} - 2, \text{iron} - 3$ |
| | scissors | $\text{at\_workbench} = 1, \text{iron} \geq 2$ | $\text{scissors} + 1, \text{iron} - 2$ |
| | paper | $\text{at\_workbench} = 1,$ $\text{scissors} \geq 1, \text{wood} \geq 1$ | $\text{paper} + 1, \text{wood} - 1$ |
| | bed | $\text{at\_toolshed} = 1,$ $\text{wood} \geq 3, \text{wool} \geq 3$ | $\text{bed} + 1, \text{wood} - 3,$ $\text{wool} - 3$ |
| | jukebox | $\text{at\_workbench} = 1,$ $\text{wood} \geq 3, \text{gem} \geq 1$ | $\text{jukebox} + 1, \text{wood} - 3,$ $\text{gem} - 1$ |
| | enhance_table | $\text{at\_workbench} = 1, \text{stone} \geq 1,$ $\text{paper} \geq 2, \text{gem} \geq 1$ | $\text{enhance\_table} + 1, \text{stone} - 1,$ $\text{paper} - 2, \text{gem} - 1$ |

Table 5: Learned ground operators of the Minecraft environment.

| Skill | Object | Preconditions | Effects |
|---|---|---|---|
| push | cube | | $\text{xy\_goal} + 1$ |
| approach | cube | $\text{is\_grab} = 0$ | $\text{is\_grab} = 1$ |
| lift | cube | $\text{is\_grab} = 1$ | $\text{xy\_goal} + 1, \text{z\_goal} + 1$ |
| press | button | $\text{next\_color} < \text{num\_objects}$ | $\text{next\_color} + 1$ |

Table 6: Learned ground operators of the IssacGym environment.

- **Color Success.** The color success gives out each color's success rate.

- **Color Success Fraction.** The color success fraction indicates the percentage of completed subtasks in each color.

- **Action Success.** The action success reflects the success rate of each action.

- **Action Success Fraction.** The action success fraction represents the percentage of accomplished subtasks in each action.

## D SUPPLEMENTARY RESULTS

### D.1 DETAILED RESULTS FOR MINECRAFT TASKS

The plans for different tasks generated by the MCTS algorithm are as follows:

- `Pickup-Mass-Grass`: move(workbench) → pickup(grass)

- `Pickup-Mass-Banboo`: move(workbench) → pickup(bamboo)

- `Make-Grass-Stack`: pickup(grass) → move(toolshed) → move(workbench) → make(grass_stack)

- `Make-Bamboo-Fence`: pickup(bamboo) → move(toolshed) → move(workbench) → make(bamboo_fence)

- `Make-Mass-Sticks`: move(workbench) → pickup(wood) → move(workbench) → make(stick) → pickup(wood) → pickup(wood) → pickup(wood) → pickup(wood) → move(workbench) → make(stick) → pickup(wood) → move(workbench) → make(stick) → make(stick) → make(stick) → pickup(wood) → move(workbench) → pickup(wood) → pickup(wood) → move(workbench) → make(stick) → make(stick) → make(stick) → make(stick) → pickup(wood) → pickup(wood) → pickup(wood) → move(workbench) → make(stick)

- `Pickup-Iron`: pickup(wood) → move(workbench) → pickup(wood) → pickup(wood) → move(toolshed) → move(workbench) → make(stick) → make(stick) → pickup(stone) → pickup(stone) → pickup(stone) → pickup(stone) → move(toolshed) → make(stone_pickaxe) → move(workbench) → pickup(iron)

- `Multiple-Goals`: pickup(stone) → move(workbench) → move(toolshed) → pickup(stone) → pickup(wood) → pickup(stone) → pickup(stone) → pickup(stone) → move(workbench) → make(stick) → pickup(wood) → move(workbench) → make(stick) → pickup(stone) → move(toolshed) → make(stone_pickaxe) → pickup(iron) → pickup(iron) → pickup(wood) → move(workbench) → make(scissors) → move(toolshed) → pickup(wool)

- `Make-Enhance-Table`: pickup(stone) → pickup(wood) → pickup(stone) → pickup(stone) → move(workbench) → make(stick) → pickup(wood) → pickup(wood) → move(workbench) → make(stick) → move(toolshed) → make(stone_pickaxe) → pickup(iron) → pickup(iron) → move(workbench) → make(scissors) → make(paper) → move(toolshed) → pickup(wool) → pickup(wood) → move(workbench) → make(stick) → pickup(stone) → pickup(wood) → pickup(wool) → pickup(wool) → pickup(wood) → pickup(iron) → move(workbench) → make(stick) → make(paper) → pickup(iron) → pickup(iron) → move(toolshed) → make(iron_pickaxe) → pickup(gem) → move(toolshed) → make(enhance_table)

### D.2 DETAILED RESULTS FOR ISSACGYM TASKS

The plans for different tasks generated by the MCTS algorithm are as follows:

- `Push-n`: push($obj_1$) → push($obj_2$) → $\cdots$ → push($obj_n$).

- `Push-Grab-n`: push($obj_1$) → push($obj_2$) → $\cdots$ → push($obj_n$) → grab($obj_n$).

- `Push-Grab-Lift`: push($obj_1$) → push($obj_2$) → $\cdots$ → push($obj_n$) → grab($obj_n$) → lift($obj_n$).

- `Ordered-Press`: press($obj^1$) → press($obj^2$) → $\cdots$ → press($obj^n$).

The superscripts of `Ordered-Press` represent that press should follow the sequence.

We list some detailed results of the tasks in IssacGym. Table 7, table 8, and table 9 demonstrate the detailed metrics of `Push`, `Push-Grab-Lift`, and `Ordered-Press`, respectively.

Additionally, we construct an experiment called `Push-Grab`. Its difficulty is between `Push` and `Push-Grab-Lift` since we expect the agent to use two skills to complete the task. The agent is required to push the cubes to their goal positions and grab one of the specified cubes. We show the detailed results under different numbers of cubes in table 10.

| Cubes | Success | Success Fraction | Color Success Fraction | Color Success | Action Success Fraction | Action Success |
|---|---|---|---|---|---|---|
| 1 | 1.000 | 1.000 | 1.000 | 1.000 | 1.000 | 1.000 |
| 2 | 1.000 | 1.000 | 1.000 | 1.000, 1.000 | 1.000 | 1.000 |
| 3 | 0.875 | 0.958 | 0.958 | 0.938, 0.938, 1.000 | 0.958 | 0.875 |
| 4 | 0.750 | 0.922 | 0.922 | 1.000, 0.875, 0.938, 0.875 | 0.922 | 0.750 |
| 5 | 0.688 | 0.900 | 0.900 | 0.938, 0.875, 0.938, 0.938, 0.813 | 0.900 | 0.688 |

Table 7: `Push`. The sequence of color success fractions follows red, green, blue, yellow, and purple.

| Cubes | Success | Success Fraction | Color Success Fraction | Color Success | Action Success Fraction | Action Success |
|---|---|---|---|---|---|---|
| 1 | 0.625 | 0.875 | 0.625 | 0.625 | 0.875 | 1.000, 0.9375, 0.688 |
| 2 | 0.500 | 0.828 | 0.719 | 0.500, 0.938 | 0.771 | 0.938, 0.875, 0.500 |
| 3 | 0.500 | 0.863 | 0.792 | 0.563, 0.938, 0.875 | 0.771 | 0.750, 0.938, 0.625 |
| 4 | 0.063 | 0.698 | 0.609 | 0.188, 0.938, 0.750, 0.563 | 0.479 | 0.438, 0.625, 0.375 |
| 5 | 0.125 | 0.643 | 0.575 | 0.3125, 0.5625, 0.5, 0.75, 0.75 | 0.438 | 0.1875, 0.625, 0.5 |

Table 8: `Push-Grab-Lift`. The sequence of color success follows red, green, blue, yellow, and purple. The sequence of action success follows push, approach, and lift.

| Cubes | Success | Success Fraction | Color Success Fraction | Color Success | Action Success Fraction | Action Success |
|---|---|---|---|---|---|---|
| 1 | 0.990 | 0.990 | 0.990 | 0.990 | 0.990 | 0.990 |
| 2 | 0.938 | 0.969 | 0.969 | 0.969, 0.969 | 0.938 | 0.938 |
| 3 | 0.875 | 0.958 | 0.958 | 0.938, 1.000, 0.938 | 0.875 | 0.875 |
| 4 | 0.813 | 0.953 | 0.975 | 0.875, 1.000, 1.000, 0.938 | 0.875 | 0.8125 |
| 5 | 0.813 | 0.950 | 0.950 | 0.875, 1.000, 1.000, 0.938, 0.938 | 0.813 | 0.875 |

Table 9: `Ordered-Press`. The sequence of color success follows red, green, blue, yellow, and purple.

| Cubes | Success | Success Fraction | Color Success Fraction | Color Success | Action Success Fraction | Action Success |
|---|---|---|---|---|---|---|
| 1 | 0.938 | 0.969 | 0.938 | 0.938 | 0.969 | 1.000, 0.9375 |
| 2 | 0.938 | 0.979 | 0.969 | 0.875, 1.000 | 0.969 | 0.938, 0.938 |
| 3 | 0.563 | 0.859 | 0.813 | 0.688, 0.875, 0.875 | 0.750 | 0.688, 0.813 |
| 4 | 0.438 | 0.838 | 0.828 | 0.688, 0.875, 0.875, 0.875 | 0.656 | 0.625, 0.688 |
| 5 | 0.250 | 0.792 | 0.763 | 0.688, 1.000, 0.688, 0.688, 0.750 | 0.531 | 0.3125, 0.75 |

Table 10: `Push-Grab`. The sequence of color success follows red, green, blue, yellow, and purple. The sequence of action success fractions follows push and approach.

## D.3 COMPOSITIONAL GENERALIZATION

For the experiments measuring compositional generalization, we provide some demonstration results in the main paper. Here, we list some of the additional results. The results are listed in Table 11 and Table 12. For the Minecraft environment, we introduce some new objects {grass, bamboo} and the corresponding crafting tasks. For IsaacGym Environment we add some new type of objects {cuboid, cylinder, star, T-block} in the environment. We can find that in most of the test cases, our model can maintain the success rate without fine-tuning the model. Also, we provide another demonstration of the experiment result in Figure 10, which is pushing the star and crafting the bamboo.

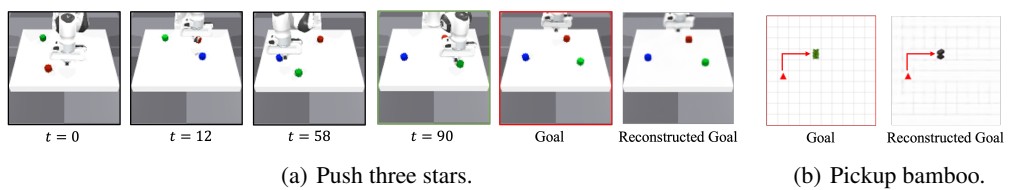

| $t=0$ | $t=12$ | $t=58$ | $t=90$ | Goal | Reconstructed Goal | Goal | Reconstructed Goal |

(a) Push three stars.  (b) Pickup bamboo.

Figure 10: Compositional generalization in IssacGym and Minecraft environments.

| Shape | Cuboid | Cylinder | Star | T-Block |
|---|---|---|---|---|
| **Ours** | 0.375 / 0.750 | 0.250 / 0.729 | 0.625 / 0.833 | 0.250/0.667 |

Table 11: Success rate of `Push` on objects with different shape.

| Tasks | Pickup-Mass-Grass | Pickup-Mass-Banboo | Make-Grass-Stack | Make-Bamboo-Fence |
|---|---|---|---|---|
| **Ours** | 0.969 | 0.927 | 0.865 | 0.791 |

Table 12: Success rate of Minecraft tasks on new materials.

### D.4 VISUALIZATION OF DLP RESULTS

We demonstrate the object reconstruction visualization of DLP in the IssacGym and Minecraft environments.

Figure 11 and figure 12 present lists of 32 images reconstructed by DLP respectively. In the Issac-Gym environment, We find that DLP focuses on objects with different colors and the gripper, while in Minecraft, object blocks and agents are clearly shown in the grid.

Figure 13 and figure 14 present a comparative analysis of the original image with various transformed versions. These include images with different key points, reconstructed images, extracted foregrounds and backgrounds, and images with different types of bounding boxes. The first row depicts the original image. Key points are marked on the original image in the second row. The third row showcases the reconstructed images, which exhibit a high degree of similarity to the originals. In the fourth row, predicted key points are superimposed on the original image, with many aligning closely with objects. The fifth row highlights the top 10 key points that the agent prioritizes, which are predominantly concentrated on meaningful objects rather than empty regions. The sixth and last rows display the extracted foregrounds and backgrounds, respectively. The foreground images effectively isolate individual objects, while the backgrounds are clean and devoid of objects. The seventh and eighth rows demonstrate the application of bounding boxes to each object using two different methods: non-maximum suppression alone and non-maximum suppression in conjunction with transparency.

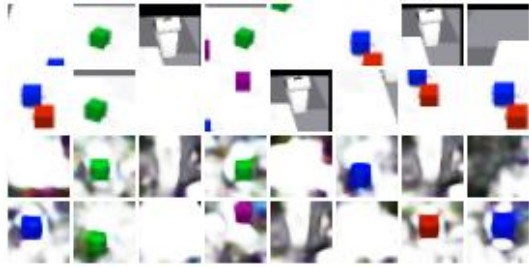

Figure 11: Object Reconstruction of DLP in IssacGym.

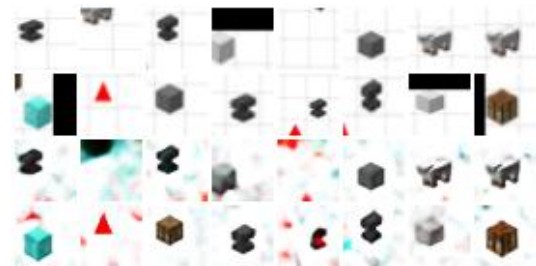

Figure 12: Object Reconstruction of DLP in Minecraft.

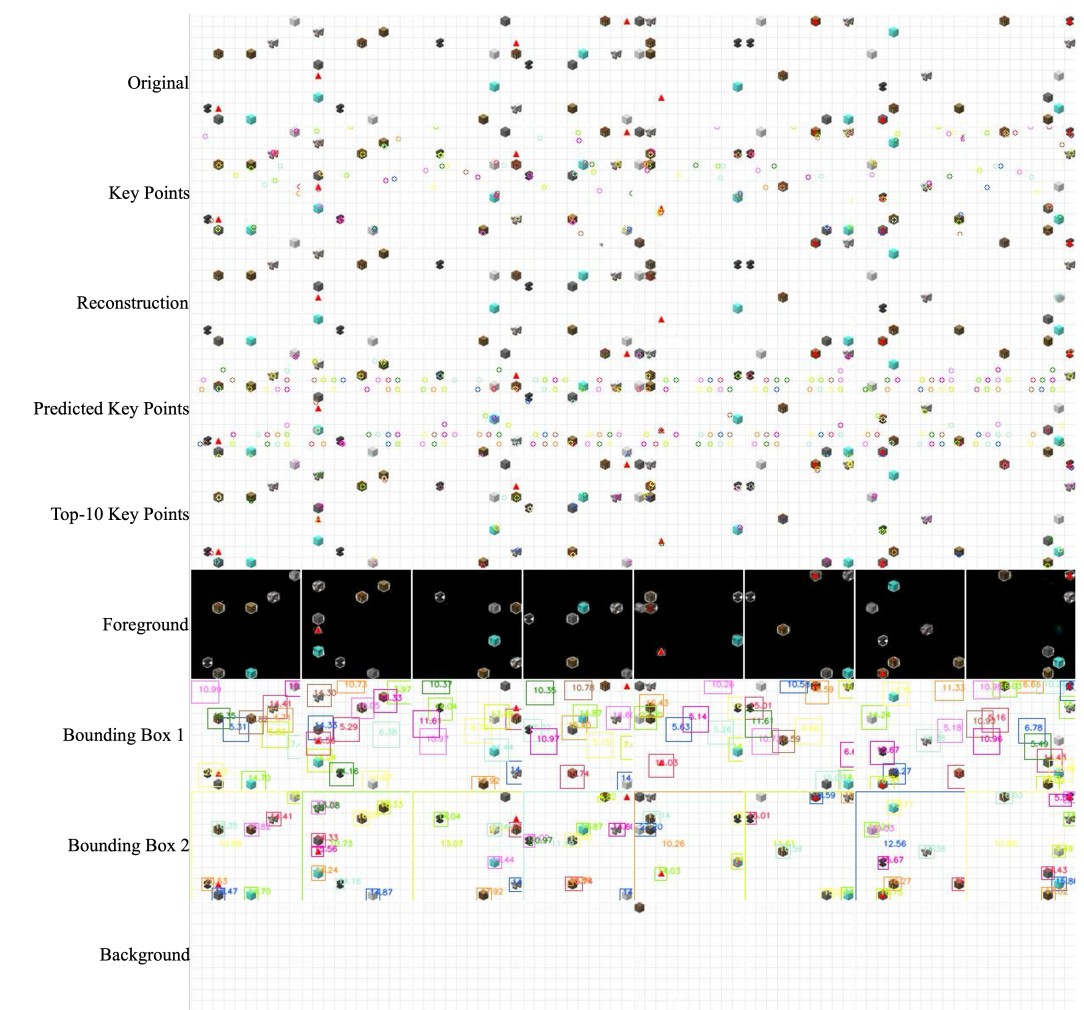

Figure 13: Visualization of DLP in IssacGym.

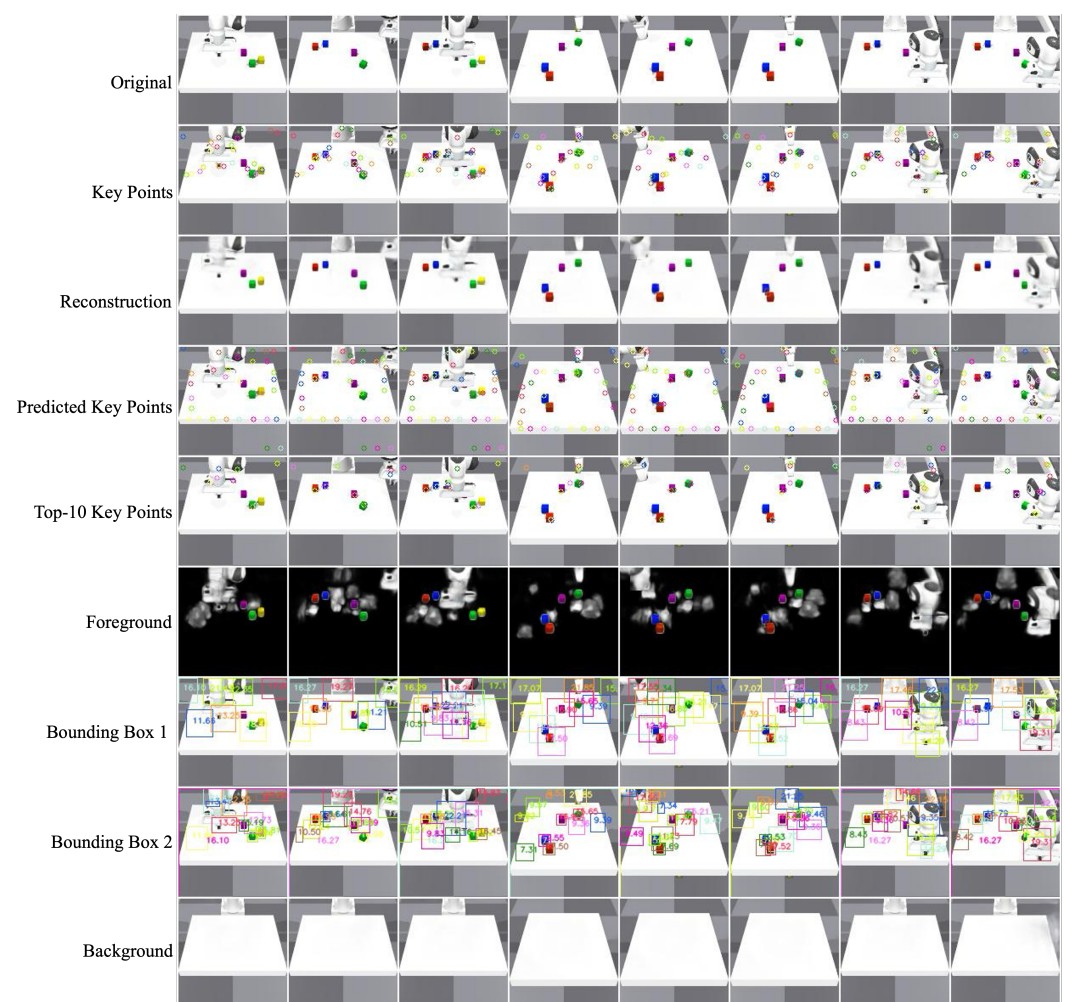

Figure 14: Visualization of DLP in Minecraft.

