# OpenReview forum: "From Skills to Plans: Automatic Skill Discovery and Symbolic Interpretation for Compositional Tasks"
_ICLR.cc/2025/Conference — ICLR 2025 Conference Withdrawn Submission_

### Official Review · Reviewer_CjfT · 2024-10-24

**Soundness:** 2
**Presentation:** 2
**Contribution:** 2
**Rating:** 3
**Confidence:** 4

**Summary:**

This work presents a method that combines several learning algorithms to acquire low level skills, high-level symbolic representations and mechanisms to bridge between the two. These are used to facilitate high-level planning to solve long-horizon pixel-based tasks that require reasoning about multiple entities and composing multiple skills.

The method is comprised of $4$ main stages:

1. Object-centric representations of state image observations are extracted using a pre-trained model. These are then further processed to produce a symbolic feature representation of the state using a Transformer-based network.
2. Individual skill policies are learned using imitation learning from data collected by a random policy that is segmented and divided into offline datasets. Learning of these policies is done on top of the OCR of images with a Transformer-based policy.
3. Preconditions of the skills are learned via neural guidance and the effects using symbolic regression.
4. A skill dependency graph is built based on the preconditions and effects and then tasks are solved with MCTS based on the initial state and goal.

The authors compare the performance of their method to reinforcement/imitation learning methods as well as methods that combine the above with planning on simulated environments that include crafting in a 2D grid world as well as tabletop multi-object manipulation.

**Strengths:**

Summarized points:
- Tackle a hard problem: long-horizon compositional tasks from pixel observations
- Seemingly interesting combination of deep reinforcement/imitation learning and planning
- Interesting use of object-centric representations for learning state abstractions
- Test method on relevant and challenging simulated environments

**Weaknesses:**

Summarized points:
- Main text is hard to follow
- Method description could be more self-contained
- Method is complex and relies on many different algorithms
- Method is not explained clearly enough and some details remain ambiguous
- Symbolic features used for planning are hand-designed, require expert knowledge of the task
- Choice of baselines is not well motivated
- Hard to make an apples-to-apples comparison with baselines due to variations in assumptions
- Lacking visualizations such as policy rollouts
- Missing an analysis of failure cases, especially of the baselines
- Reproducibility: No code, lacking many details and hyper-parameters

**Intro**

“They cannot simultaneously learn diverse skills due to the catastrophic forgetting problem” - I think some more recent and directly relevant citations are missing here to back up this claim. There have been large models that are able to learn diverse skills even across tasks and embodiment with large enough networks and appropriate architectural mechanisms, e.g. [TD-MPC2](https://arxiv.org/abs/2310.16828).

*Automatic Skill Discovery Contribution*: “Without any guidance of designed high-level symbolic representations in advance” - it seems to me that this is not the case with your method. For example, from my understanding, the traces from which the skills are learned are segmented based on the feature vectors which are hand-designed per environment in advance.

**Skill with Symbolic Interpretation**

I am missing a definition of $pre_l(s,f)$ and $eff_l(s,f)$ in comparison to the corresponding ground operator precondition and effects. What is their output?

Why do the above functions depend both on $s$ and $f$, could you possibly provide an example from the simulated environments to explain this point?

Shouldn’t $\pi_l$ be goal-conditioned in your framework?

**Method - Feature Extraction**

To my understanding, the feature state of the environment is hand-designed and requires access to the ground-truth state of the environment during training. This should be mentioned explicitly in the main text. It requires expert knowledge of the environment and in itself solves a significant part of the problem by defining the aspects of the states and attributes of interest for solving the task.

The role of the cross-attention in the aggregation Transformer is not clear to me.
Cross-attention is an operation between $2$ sets, one is the state entities and what is the other? What does “temporal difference” mean in the context of the current state entities?

**Method - Skill Learning**

*Trace Generation*:  What is a significant number? What are the lengths of the traces? These details are not in the appendix and I think there should be some comparison to the baselines in terms of sample efficiency.

*Trace Categorization*: this paragraph is not clear.
- What does a feature of the trace mean? Features are defined per-state while traces are sequences of states and actions as defined in Section 4.
- The segmentation procedure is not clear and the motivation for the segmentation is not explained. Maybe an explicit example here would help.
- How do you measure the L2 distance between traces? Why is this a good measure for your purposes? How do you choose the number of clusters $K$?

*Training the Individual Skill Policies with GCBC*:
- What is the role of the first term in the loss? As I understand it, this loss is meant to train the skill policy parameters which the first term has nothing to do with. Also, what does subscript $t$ mean in the first term? Should it be $T$?
- Is the skill policy deterministic? How does it handle the diversity of trajectories in each offline dataset?
- Since the skills are a result of behavioral cloning of a random policy, does that not produce inefficient skills?

**Method - Symbolic Inductive Inference**

What is the ‘complexity’ term in the symbolic regression loss explicitly? When you refer to it as a ‘normalization’ term do you mean regularization?

I think this part should be more self-contained and include more examples from your experimental environments to help understand what exactly you want to obtain with each algorithm described here.

**Method - End-to-End Pixel-Based Planning**

What is the purpose of the image segmentation? What does it mean to segment an image according to its semantics? What algorithm do you use to do this? This seems like a key aspect in the implementation of your method that is not described or visualized in any part of the paper.

It is not clear how you use a skill learned in a simple environment, e.g. a policy trained on images of a single cube in IsaacGym, and apply it to more complex environments, e.g. to images with multiple cubes.

**Experiments - Baselines**

The experiment section lacks a motivation for the choice of baselines. Some questions I think should be answered are:
- How is each baseline directly comparable to your method? What assumptions does each method make?
- What is the SOTA on each of the environments in your experiments?
- How does each baseline help you answer the questions you want to answer with your experiments?

I find that many details are missing about the baselines:
- What are the rewards that the RL baselines (SMORL, ECRL) are trained on in each environment?
- What imitation data is GAIL trained on and how is it acquired?
- Can you add a more explicit comparison between your method’s pipeline and the other two methods that combine planning and DRL? Possibly in the appendix.

**Experiments - Long-Horizon Sequential Task**

What are the simple environments the agent is trained in to acquire the basic skills? Do you have to design a simple-environment-suite that contains all the individual tasks an agent may need to solve as part of a longer sequence?

What do you mean by the fact that SMORL/ECRL cannot handle temporal logic tasks? I would expect that if you are able to learn the individual skills needed to solve the long-horizon tasks from random exploration data, it shouldn’t be too hard for an RL agent to learn the same behaviors from an appropriately designed reward. What are the failure cases?

Why does GAIL fail? I would expect imitation learning from expert data to be relatively straightforward in this task.

**Experiments - Object Manipulation**

How are the RL and IL baselines trained on an increasing number of cubes? Are they trained from scratch in each environment?

I do not understand the claim that Deepsynth and DiRL performs poorly on increasing number of objects because their decompositional logic is simple and relies on expert knowledge. What in their simplicity hinders their performance? What expert knowledge do they use that you do not? I think the analysis of failure cases can also help strengthen your point.

I am not sure I agree with the claim that the awareness of the temporal attributes of sub-tasks demonstrates the superiority of your method over the RL baselines in this case. Your method uses an explicit mechanism for enforcing the order of subtasks by specifying the state variables ‘color’ and ‘next_color’. If a similar notion of ordering would be expressed in the reward of the RL baselines, the agent could optimize through learning to achieve the goal in a specific order just like your method optimizes through planning for a specific order. My main point here is that the goal specification in the RL methods does not contain all the information that you use to define success in the task, while it does for the planning methods.

**Experiments - Compositional Generalization**

What is your definition of compositional generalization?

What aspects does the agent need to compose zero-shot in order to solve the new tasks?

Why are the generalization results in the modified environments not trivial? It seems to me that the objects are changed in a way that does not change the actual task or the skills required to solve them and only very slightly changes their appearance.

**Questions:**

For major questions and requests/remarks, see weaknesses.

Additional minor questions/remarks:
- I believe the term “real-world” should be reserved for environments that are in our physical world and not in simulation. I suggest replacing “real-world” with “simulated” when referring to the IsaacGym object manipulation environments.
- Related Work: [FOCUS](https://arxiv.org/abs/2307.02427), [DAFT-RL](https://arxiv.org/abs/2307.09205), [HOWM](https://arxiv.org/abs/2204.13661) are all model-based algorithms, not model-free.
- It would be helpful for the reader to have references in the method section in the main text to the Appendix containing the relevant materials and additional explanations.

---

### Official Review · Reviewer_Ufge · 2024-10-29

**Soundness:** 1
**Presentation:** 1
**Contribution:** 2
**Rating:** 3
**Confidence:** 4

**Summary:**

This work proposes a pipeline that:
1. first clusters action sequences into skills,
2. learns symbolic precondition and effect functions for those skills that represent the necessary conditions to execute a skill, and what happens when executed,
3. and generates plans to arrive at a goal state—featurized and represented by a neural network.
In a high-level, it can be thought of as a model-based method where we learn action sequences, i.e., skills, learn preconditions and effects for those skills, and since this model is symbolic, it affords graph-based search to plan for a goal state. Experiments done in a 2D Minecraft domain and a tabletop object manipulation domain show that the method performs better than the compared baselines.

While the paper might be in the interest of the community, I think it needs at least one more iteration to strengthen its motivation by removing the ambiguities in the text and by covering the related works and how this method stands apart from them to clearly show its core contribution.

**Strengths:**

- While there are some other works in a similar spirit (will be discussed below), this work sets itself apart from others by using symbolic regression to learn precondition and effect sets, which might be a promising alternative.
- The second strong point is the focus on skill learning as well. Methods that focus on state abstraction mostly assume the existence of high-level skills. While there is merit in doing so, focusing on both problems at the same time also is a nice positioning of the overall problem, at the cost of making the whole pipeline harder.
- There is a nice set of experiments with multiple baselines. There are also some nice outputs of the method in the appendix regarding the structure of the skill. I would recommend maybe showing some exemplary outputs also in the main manuscript.

**Weaknesses:**

# Lack of reference to fundamental works in this area
It is surprising a work named “From Skills to Plans: …” misses a very fundamental work in this area by Konidaris et al. [1], named “From Skills to Symbols: …”, which won the IJCAI-JAIR 2020 best paper award. In that work, they derive an algorithm to learn provably necessary and sufficient symbolic representations of the environment, and use these symbols to do long-horizon planning. Its object-oriented counterpart applies this algorithm in domains like Minecraft (both in 2D and from pixels in 3D) [2], which is even more related considering this paper also uses the 2D Minecraft domain. And there are world-model-based works from the RL community including but not limited to [3, 4, 5, 6], and see [7].

> Line 50, Many existing methods (Illanes et al., 2020; Sun et al., 2020; Zhuo et al., 2021; Mao et al., 2023; Silver et al., 2023) employ a top-down structure by specifying symbolic representations for high-level action models and using them to guide the learning of low-level policies.

There are works including but not limited to [1, 8, 9, 10, 11, 12] that learn symbolic abstractions in a bottom-up fashion.

> Line 90, Object-centric RL …

Diuk et al. [13] is an early work that defines object-oriented MDPs.

> Line 78, Automatic skill discovery

There is no mention of other works that focus on automatic skill discovery. Off the top of my head, [14, 15] learns a graph of skills from pixels in complex environments, and I believe there should be more in the context of hierarchical RL.

UCT [18]

# Clarity
While the overall idea makes sense—learning preconditions and effects of skills, and chaining them to plan, some of the definitions and the motivation on the use of specific methods is lacking:
- It’s not clear why we start from the object-centricness assumption. Since we’re using neural nets to learn the representation, and the output, which is defined as the feature state (?), is a fixed-size vector as opposed to representing the environment as a set of symbols for each object, there is no gain on the planning part due to having an object-factored environment.
- The second important point is regarding the use of symbolic inductive inference (Sec. 5.3). On Line 267, “To form a plan using the skills, we need to get a symbolic interpretation of the skill”, it’s not mentioned why exactly we would want symbolic interpretations. The actor in AlphaGo [17] does not have any symbolic interpretation, yet, is used with MCTS to plan-ahead. Furthermore, it’s not motivated why we would use such mathematical operations and do symbolic regression. It’s not that these don’t make sense (which I think they might do), but rather the specific reason to use such methods instead of others. It would be beneficial for readers to understand why such methods are chosen, so that they can speculate on what other candidate methods could be used, too.
- I didn’t understand why we need the dependency graph. Isn’t the symbolic definition of the skill give us a way to do tree search? Why construct the graph explicitly?
- Line 114, “we propose a novel approach to bridge the gap between MDP and planning.” What is this gap?
- Line 131, do we implicitly assume that object-factorization still retains the Markov property of the environment?
- Definition 2, in Diuk et al. [13], this is defined as OO-MDP.
- Definition 4, it’s not clear how eff(o) and \beta are different. Also, quite strange to miss the semi-MDP work [16] where they define the option tuple, but somehow use the same notation for the termination set, \beta.
- In Definition 5, I didn’t quite understand why we would need state in the tuple. It feels like an argument of the policy.
- Line 214, “m indexing the number of factorization”, what is the number of factorization, or what is factorization? I believe it’s not object-factors as N is the number of entities.
- In Sec. 5.4, “we use an image segmentation algorithm to segment the image according to its semantics”, this is not mentioned before, and not clear why we are using the segmentation.
- In Table 2, “Success rate and success fraction of real-world object manipulation.”, but there is no detail on the real-world object manipulation environment.

# References
1. Konidaris, George, Leslie Pack Kaelbling, and Tomas Lozano-Perez. "From skills to symbols: Learning symbolic representations for abstract high-level planning." Journal of Artificial Intelligence Research 61 (2018): 215-289.
See the best paper award, 2020: https://www.jair.org/index.php/jair/IJCAIJAIR
2. James, Steven, Benjamin Rosman, and G. D. Konidaris. "Autonomous learning of object-centric abstractions for high-level planning." Proceedings of the The Tenth International Conference on Learning Representations. 2022.
3. Ha, David, and Jürgen Schmidhuber. "World models." arXiv preprint arXiv:1803.10122 (2018).
4. Hafner, Danijar, et al. "Dream to control: Learning behaviors by latent imagination." arXiv preprint arXiv:1912.01603 (2019).
5. Hafner, Danijar, et al. "Mastering atari with discrete world models." arXiv preprint arXiv:2010.02193 (2020).
6. Hafner, Danijar, et al. "Mastering diverse domains through world models." arXiv preprint arXiv:2301.04104 (2023).
7. Matsuo, Yutaka, et al. "Deep learning, reinforcement learning, and world models." Neural Networks 152 (2022): 267-275.
8. Ugur, Emre, and Justus Piater. "Bottom-up learning of object categories, action effects and logical rules: From continuous manipulative exploration to symbolic planning." 2015 IEEE International Conference on Robotics and Automation (ICRA). IEEE, 2015.
9. Ahmetoglu, Alper, et al. "Deepsym: Deep symbol generation and rule learning for planning from unsupervised robot interaction." Journal of Artificial Intelligence Research 75 (2022): 709-745.
10. Asai, Masataro, et al. "Classical planning in deep latent space." Journal of Artificial Intelligence Research 74 (2022): 1599-1686.
11. Chitnis, Rohan, et al. "Learning neuro-symbolic relational transition models for bilevel planning." 2022 IEEE/RSJ International Conference on Intelligent Robots and Systems (IROS). IEEE, 2022.
12. Shah, Naman, et al. "From Reals to Logic and Back: Inventing Symbolic Vocabularies, Actions and Models for Planning from Raw Data." arXiv preprint arXiv:2402.11871 (2024).
13. Diuk, Carlos, Andre Cohen, and Michael L. Littman. "An object-oriented representation for efficient reinforcement learning." Proceedings of the 25th international conference on Machine learning. 2008.
14. Bagaria, Akhil, and George Konidaris. "Option discovery using deep skill chaining." International Conference on Learning Representations. 2019.
15. Bagaria, Akhil, Jason K. Senthil, and George Konidaris. "Skill discovery for exploration and planning using deep skill graphs." International Conference on Machine Learning. PMLR, 2021.
16. Sutton, Richard S., Doina Precup, and Satinder Singh. "Between MDPs and semi-MDPs: A framework for temporal abstraction in reinforcement learning." Artificial intelligence 112.1-2 (1999): 181-211.
17. Silver, David, et al. "Mastering the game of Go with deep neural networks and tree search." nature 529.7587 (2016): 484-489.
18. Kocsis, Levente, and Csaba Szepesvári. "Bandit based monte-carlo planning." European conference on machine learning. Berlin, Heidelberg: Springer Berlin Heidelberg, 2006.

**Questions:**

Copied from the above part:
- I didn’t understand why we need the dependency graph. Isn’t the symbolic definition of the skill give us a way to do tree search? Why construct the graph explicitly?
- Line 114, “we propose a novel approach to bridge the gap between MDP and planning.” What is this gap?
- Line 131, do we implicitly assume that object-factorization still retains the Markov property of the environment?
- In Definition 5, I didn’t quite understand why we would need state in the tuple. It feels like an argument of the policy.
- Line 214, “m indexing the number of factorization”, what is the number of factorization, or what is factorization? I believe it’s not object-factors as N is the number of entities.
- In Sec. 5.4, “we use an image segmentation algorithm to segment the image according to its semantics”, this is not mentioned before, and not clear why we are using the segmentation.
- In Table 2, “Success rate and success fraction of real-world object manipulation.”, but there is no detail on the real-world object manipulation environment.

---

### Official Review · Reviewer_voYo · 2024-10-30

**Soundness:** 2
**Presentation:** 2
**Contribution:** 2
**Rating:** 3
**Confidence:** 4

**Summary:**

Paper propose to combine object-centric representations with symbolic representations for learning complex long-term tasks. In particular, the authors propose to first learn skills in simple environments and then combine them by searching in the space of skills mapping to symbols. The authors show that such approach allows for compositional generalization by applying skill to a novel object with similar features.

**Strengths:**

- Overall problem of planning to tackle complex compositional problems that require both learning skills to control independent parts of the environment and to combine those to tackle complex problems. The paper proposes to map object-centric representations and skills on top of them into symbolic space where planning is possible.
 -  The idea of mapping the representations to some describe set of "features" to simplify planning is interesting. While discovery of such features is challenging, potentially in the future some VLM methods could be used to provide discrete feature spaces

**Weaknesses:**

- Skill Learning & Trace Generation: there is assumption to have access to simple environments. However, in the real-world this assumption is not achievable as for this one need to already discover structure and isolate the objects. Also, other baselines are not assuming it, thus it would be great to see how the method in supposed to be used for the real-world environments.

- OCR is first learned, but then aggregated to some feature state. Why do we need object-centric representation in the first place, if we aggregate it further? In other words, is it possible to map from image observations to the feature state directly?

- Given the large number of components, it is not clear which of them are important and how they are related. Provide an extensive ablation study including but not limiting to the importance of object centric representations, simple environments for skill learning, using of EQL as prediction rule and so on.

- Usage of ground-truth masks and overall simplicity of the visual representations, making claim about end-to-end pixel based planning overstatement. It would be great, if the authors can showcase their methods for visually more challenging environments, for example by using DINOSAUR [1] or VideoSAUR [2] for scaling to more challenging multi-object environments while not using GT masks. Alternatively, one can use some foundational models like Grounded-SAM[3] to segment objects and extract some representations. Overall, it is crucial that similar approach is applicable for more realistic environments while not assuming that environment structure in terms of segmentation masks is given.

- Some parts of the paper is difficult to read. For example, Figure 1 could be separated to parts while covering more details on how exactly we map from object-centric representions to features then to skills and finally how we combine skills.
[1] https://arxiv.org/abs/2209.14860

[2] https://arxiv.org/abs/2306.04829

[3] https://github.com/IDEA-Research/Grounded-Segment-Anything

**Questions:**

### Clarification questions :
- Is $T_f$ learned or given? From Ef 1 it seems life some f_i is given and another is estimated (f_i). So please clarify, how do we obtain $f_i$ and what is assumed to be provided by the environment vs what is discovered or learned by agents.

- “In the above definition, the precondition prel(s, f ) and effect eff l(s, f ) are both a function of the input state and feature, which means there are multiple legal feature states for a particular skill. “ Not clear what is the meaning for multiple legal feature states

- “Another possible direction is to employ generative models, such as diffusion models, to replace the current image segmentation approach to generate sub-goal images” not clear how generation of the subgoals it connected with segmentation.

Small fixes:

093: 1. Francesco et al., 2020 -> Locatello et al., 2020

246: “k-means clustering algorithm (Ahmed et al., 2020)” it is better to cite original work not survey

---

### Official Review · Reviewer_F7o7 · 2024-11-04

**Soundness:** 2
**Presentation:** 1
**Contribution:** 2
**Rating:** 3
**Confidence:** 3

**Summary:**

This paper proposes a method for solving goal-conditioned, pixel-based control tasks that require long-horizon reasoning and compositional generalization.  First, random rollouts are executed on a simplified version of the task.  Next, the rollouts are clustered into different datasets based on changes in the feature representation during the rollout.  Each dataset is meant to represent a different skill that the agent may use.  To instantiate each skill, the dataset is used to train a pre-condition function (indicates when a skill should be used), an effect function (indicates how a skill changes the state or representation of the environment), and a policy function (indicates what low-level actions to perform during the skill).  During evaluation, MCTS is used to find a sequence of skills to take the agent from the start state to the goal state.  To facilitate compositional generalization, an object-centric representation is first extracted from the image observations before feeding to the skill components.  The method is evaluated on two environments, a grid-world version of Minecraft and a robotic manipulation setup in IsaacGym.  The method outperforms existing baselines, especially on longer-horizon tasks.

**Strengths:**

- The paper introduces a method that is able to learn useful skills from random rollout data.  Distinct skills are extracted by clustering rollouts based on changes in the feature representation between states.
- The paper provides an approach for training policies, pre-condition and effect functions that define each skill, such that MCTS can be used to search for sequences of skills that achieve a desired goal state.
- The method is shown to outperform competitive baselines on gridworld Minecraft and IsaacGym manipulation tasks.  The method significantly outperforms the baselines when the task complexity is increased.

**Weaknesses:**

- The notation is not consistent or imprecise.  Here are some examples: Section 3.1 Definition 2, $\mathcal{P}_m$ is used without stating what it is.  Definition 5 is confusing: why is the tuple $l$ a function of $\mathbf{s}$? Why is the ground operator included in the tuple but also the precondition and effect, which are components of the ground operator?  In Section 5.4, you introduce "subgoal state $g^i$" without ever saying how a subgoal is generated.
- Some details are missing or not clearly stated.  For instance, in Equation (1), you introduce $\hat{f}$, which presumably means ground truth features?  Where do these ground truth features come from?  A brief statement in Appendix C ("we design the features to extract as follows:") makes it sound like these are hand-designed using access to the simulator state.  If so, how would such features be learned in real world environments?
- More details are needed to demonstrate that the comparison to baselines was fair.  The proposed method makes use of random rollouts in a simplified environment.  Do the baselines rely on the same data or different data?
-  In Definition 4, you introduce the termination condition of the ground operator is discussed as $\beta \in \mathcal{F}_g$.  But in practice you "set the policy a time horizon as *t*" (top of page 7).  This is confusing, and no explanation is offered as to why the method did not train a termination condition function.
- Please add error bars to the results in Table 1 and Table 2 and state how many seeds were used.
- There is no ablation study to understand what aspects of the proposed method contribute to its performance.  This is a serious weakness of the paper, given the method is quite involved.  Possible ablations that would be of interest: swapping the OCR model with a standard image feature extractor, swapping the symbolic regression  method (PySR) with a traditional regression network, swapping the precondition network (EQL) with simple binary classification network, or using images directly instead of running a segmentation model.  '
- There are lots of tunable parameters to the method (including the time horizon limit of the skill policies, the max number of entitities, number of skills, etc).  Could you add some experiments that demonstrate the sensitivity of the method to different values of these parameters?
- The proposed method relies on extracting useful skills from random rollouts in a simplified environment, and a convincing case is not made that the method would be successful on more challenging tasks without significant engineering of this "simplified" environment.
- Some wording is a bit awkward.  Page 7: "A basic experiment that **urges** the agent to produce a stick" (requires?).  Page 7: "Press different buttons in an **inherent** order" (specified? or particular?).  Page 8: "Here, we **mainly demonstrate** the ..." (report?).  Page 8. "**Contrarily**" (In contrast?).  Page 9: "Isaacgym" (IsaacGym).

**Questions:**

- Based on Algorithm 1, the method is only trained on the random rollout data.   Is there any ability for the method to be improved using on-policy data?  One could imagine a scenario where a skill is not perfectly learned, and refinement is needed to successfully link it with other skills.
- Based on the explanation at the end of Section 4, the precondition function takes $s$ and $f$ as input, but the effect function only predicts the change to $f$.  This would imply that the effect function is not sufficient to perform MCTS (how do you know the resulting state after performing a skill).  Could you please clarify this?
- In Equation 2, the clustering is performed based on the L2 distance between traces.  How does this work?  Is a trace represented as the concatenation of features for each time step? What about misalignement in time?  Please provide a more precise definition of a trace (the current one at the end of Section 4 is unclear).

---

### Note · Authors · 2024-11-18

I have read and agree with the venue's withdrawal policy on behalf of myself and my co-authors.